# Trends in China's anthropogenic emissions since 2010 as the consequence of clean air actions

Bo Zheng[1,†], Dan Tong[2], Meng Li[2], Fei Liu[1], Chaopeng Hong[2], Guannan Geng[2], Haiyan Li[1], Xin Li[2], Liqun Peng[1], Ji Qi[1], Liu Yan[2], Yuxuan Zhang[2], Hongyan Zhao[2], Yixuan Zheng[2], Kebin He[1,2], and Qiang Zhang[2]

[1]State Key Joint Laboratory of Environment Simulation and Pollution Control, School of Environment, Tsinghua University, Beijing 100084, People's Republic of China
[2]Ministry of Education Key Laboratory for Earth System Modeling, Department of Earth System Science, Tsinghua University, Beijing, China
[†] Present address: Laboratoire des Sciences du Climat et de l'Environnement, CEA-CNRS-UVSQ, UMR8212, Gif-sur-Yvette, France

*Correspondence to*: Qiang Zhang (qiangzhang@tsinghua.edu.cn)

**Abstract.** To tackle the problem of severe air pollution, China has implemented active clean air policies in recent years. As a consequence, the emissions of major air pollutants have decreased and the air quality has substantially improved. Here, we quantified China's anthropogenic emission trends from 2010–2017 and identified the major driving forces of these trends by using a combination of bottom-up emission inventory and Index Decomposition Analysis (IDA) approaches. The relative change rates of China's anthropogenic emissions during 2010–2017 are estimated as follows: −62% for $SO_2$, −17% for $NO_x$, +11% for NMVOC, +1% for $NH_3$, −27% for CO, −38% for $PM_{10}$, −35% for $PM_{2.5}$, −27% for BC, −35% for OC, and +16% for $CO_2$. The IDA results suggest that emission control measures are the main drivers of this reduction, in which the pollution controls on power plants and industries are the most effective mitigation measures. The emission reduction rates markedly accelerated after the year 2013, confirming the effectiveness of China's Clean Air Action that was implemented since 2013. We estimated that during 2013–2017, China's anthropogenic emissions decreased by 59% for $SO_2$, 21% for $NO_x$, 23% for CO, 36% for $PM_{10}$, 33% for $PM_{2.5}$, 28% for BC, and 32% for OC. NMVOC emissions increased and $NH_3$ emissions remained stable during 2010–2017, representing the absence of effective mitigation measures for NMVOC and $NH_3$ in current policies. The relative contributions of different sectors to emissions have significantly changed after several years' implementation of clean air policies, indicating that it is paramount to introduce new policies to enable further emission reductions in the future.

## 1 Introduction

China produces the most air pollution in the world and contributes 18–35% of global air pollutant emissions (Hoesly et al., 2018). The major air pollutants China emits the most include sulfur dioxide ($SO_2$), nitrogen oxides ($NO_X$), carbon monoxide (CO), nonmethane volatile organic compounds (NMVOC), ammonia ($NH_3$), and particulate matter (PM), including black

carbon (BC) and organic carbon (OC). These pollutants constitute the majority of the precursors of $PM_{2.5}$ and $O_3$ pollution as well as those of short-lived climate forcers, which exert harmful effects on human health, agriculture, and regional climate. These pollutants not only cause local to regional environmental problems such as premature deaths and agricultural yield losses but also have a significant impact on regional climate changes in temperature and precipitation. To tackle the problems of both air pollution and regional climate change, it is important to fully understand the trends and drivers of Chinese emissions.

The years since 2010 have been an extraordinary period for China in the fight against air pollution. For the first time, China has added the index of $PM_{2.5}$ into its air quality standards, with an annual upper mean limit of 35 μg m$^{-3}$ (Zhang et al., 2012). In 2013, the annual average concentrations of $PM_{2.5}$ were 106 μg m$^{-3}$, 67 μg m$^{-3}$, and 47 μg m$^{-3}$ in Beijing-Tianjin-Hebei, the Yangtze River Delta, and the Pearl River Delta, respectively; these concentrations are all worse than China's 35 μg m$^{-3}$ standard and five to ten times higher than the WHO's $PM_{2.5}$ guideline value of 10 μg m$^{-3}$ (World Health Organization, 2006). To attain this air quality standard, China has strengthened its emission standards to achieve reductions in air pollutant emissions. These upgraded emission standards and the timeline for their implementation have accelerated since 2013 when the Action Plan on the Prevention and Control of Air Pollution (denoted as the Clean Air Action) was implemented (China State Council, 2013). The Clean Air Action is China's first five-year plan (2013–2017) that radically tightened air pollution targets for particulate matter pollution reduction. The three metropolitan regions mentioned above were required to reduce $PM_{2.5}$ concentrations by 15–25% by the year 2017 compared with the 2013 levels, and all other provinces in China were required to reduce $PM_{10}$ concentrations by 10%. The Clean Air Action launched stringent measures to achieve these air quality targets, including the adjustment of energy mix and industrial structure, the reduction of air pollutant emissions, the establishment of monitoring and early-warning systems for air pollution, and other supportive policies. With the successful policy implementation, China met the 2017 air pollution target set under 2013 Clean Air Action, and the annual average $PM_{2.5}$ concentrations reduced by 28–40% from 2013–2017 in the three metropolitan regions (China, 2018). Space- and ground-based observations have also confirmed the improvement of China's air quality (Krotkov et al., 2016; Liu et al., 2016; Zhang et al., 2017; Zhao et al., 2017; Zheng et al., 2018).

Establishing linkages between air quality improvements and mitigation efforts requires the use of the most recent emission inventory. However, there are no official data about how much air pollutants are emitted by China every year. The inventories developed by researchers often lag several years behind the present, leaving China without up-to-date emission inventories. Currently, there are no emission datasets that cover the period of 2010–2017. To understand the progress in air cleaning, we are in urgent need of China's most recent emission inventories, which will benefit both scientific studies and policy-making. Given that China accounts for approximately one-third of global emissions, these data will also facilitate a better understanding of the latest trends in global emissions.

In this paper, we analyze the key trends and drivers of China's anthropogenic emissions from 2010–2017. During this period, China announced unprecedented measures to improve air quality. The purpose of this study is to summarize what China has done in recent years and to evaluate how these actions have influenced anthropogenic emission trends. We first provide a comprehensive overview of China's clean air actions since 2010, especially the stringent measures that took effect after 2013

(Sect. 2). Then, we use a bottom-up method (Sect. 3) to estimate the 2010–2017 trend in Chinese emissions (Sect. 4.1) shaped by these mitigation measures. The driving factors are analyzed at the national and sectoral levels using the approach of Index Decomposition Analysis (Sect. 4.2). We separate the influence of pollution control from the influence of economic growth on the emission trend. Finally, the emission trends are evaluated against space- and ground-based observations of $SO_2$, $NO_2$, and

$PM_{2.5}$, as well as top-down constraints inferred from these observations (Sect. 4.3). Concluding remarks are given in Sect. 5.

## 2 China's clean air actions

The clean air policies that have been implemented by China since 2010 are summarized in Fig. 1. These mitigation measures cover all the major source sectors and have become increasingly stringent over time. Before 2013, strengthening the emission standards for power and industrial sectors was the key pollution control measure. For example, the emission limits of coal-

fired power plants were 400 mg m$^{-3}$ for $SO_2$, 450–1100 mg m$^{-3}$ for $NO_x$, and 50 mg m$^{-3}$ for particulates before 2012 (the standard GB 13223-2003). After 2012, all new and existing coal-fired plants were required to achieve new limit values of $SO_2$, $NO_x$, and particulates of 100 (200 for existing units), 100, and 30 mg m$^{-3}$ (the standard GB 13223-2011), respectively. China also set new emission standards for the flat glass industry and the iron and steel industry before 2013. Because other industries (e.g., the cement industry and industrial boilers) lacked stringent emission standards, they still used outdated legislation on

emission limits implemented approximately ten years ago during the period of 2010–2013.

China committed to reducing $PM_{2.5}$ pollution in 2013 for the first time ever. To fulfill the air quality target set under 2013 Clean Air Action, the government developed eight pollution control measures that were more stringent and ambitious than ever before. With these new measures, the emission limits set by existing standards were further tightened, and more stringent emission source controls were adopted not only to reduce emissions but also to improve energy efficiency and promote a

structural change in energy use patterns. We briefly describe the eight measures implemented during 2013–2017 in the following section.

1) "Ultralow" emission standard for power plants. Strengthening emission standards is key in the pollution control of coal-fired power plants. With the 2012 emission standard enacted and fully met, China pledged in December 2015 to further reduce emissions from coal power by 60% by 2020 using the "ultralow emission" technique. The emission limits for $SO_2$, $NO_x$, and

particulates are 35, 50, and 10 mg m$^{-3}$, respectively, which means that emissions from coal-fired plants must be brought to the level of those from gas-fired plants. Of the current power plants, 71% operated close to "ultralow emission" levels in 2017 (China, 2018). This figure is estimated on the basis of firm-level information of pollution control devices and efficiencies, which are collected from each plant by local agencies, and then managed and verified by Ministry of Ecology and Environment in China. The power plants that comply with "ultralow emission" standards are mainly large ones at the current stage. Most of

them use continuous emission monitoring systems to monitor exhaust emissions, which confirm that these plants are indeed complying with the "ultralow" emission levels.

2) Phase out outdated industrial capacity. Small and inefficient factories that cannot meet efficiency, environment, or safety standards have been eliminated in recent years. As a result, the average energy intensity, or energy consumed per unit of industrial gross output, steadily decreased for steel (−3.3%), cement (−2.9%), aluminum (−1.0%), ethylene (−4.2%), and synthetic ammonia (−3.0%) from 2013–2016 (National Bureau of Statistics, 2018a). The average efficiency of coal-fired power units, or grams of coal equivalent consumed per kilowatt-hour of the power supply, improved from 321 gce kWh$^{-1}$ to 309 gce kWh$^{-1}$ from 2013–2017 (National Bureau of Statistics, 2018a; National Energy Administration, 2018).

3) Strengthen industrial emission standards. Since 2013, all industrial emission standards have been strengthened; limits have been tightened and the targeted emission sources and air pollutants have been expanded. Figure 1 summarizes all the national emission standards specific to air pollutants, and there are another 22 comprehensive standards that specify the maximum amounts of waste materials in gas and water for different industries. These standards, including both new and upgraded ones from previous levels, have covered all the emission-intensive industries in China, including iron and steel making, cement, brick, coke, glass, and chemical industries. For example, the emission limits of cement plants were 800 mg m$^{-3}$ for $NO_x$ and 50 mg m$^{-3}$ for particulates before 2014 (the standard GB 4915-2004), while after 2014 all cement plants were required to reach new limit values of 400 mg m$^{-3}$ for $NO_x$ and of 30 mg m$^{-3}$ for particulates (the standard GB 4915-2013). For coal boilers used in industries, the emission limits were 900 mg m$^{-3}$ for $SO_2$ and 80–250 mg m$^{-3}$ for particulates before 2014 (the standard GB 13271-2001), and no limits were required for $NO_x$. After 2014, new coal-fired industrial boilers faced stricter limit values of 300, 300, and 50 mg m$^{-3}$ for $SO_2$, $NO_x$, and particulates (the standard GB 13271-2014), respectively. The new emission standard also tightened the limit values for existing coal-fired industrial boilers, where the "not to exceed" limits for $SO_2$, $NO_x$, and particulates were 400, 400, and 80 mg m$^{-3}$, respectively.

4) Phase out small, high emitting factories. The tightened emission standards have driven industries to upgrade and adopt cleaner technologies. Small and polluting factories that cannot meet emission standards are being retired and replaced with larger facilities. Large industrial plants that have the potential to improve their overall performance by upgrading are required to meet the latest emission legislation as early as possible. These plants must operate close to or better than the original design performance and install end-of-pipe pollution control devices to reduce emissions.

5) Install NMVOC emission control facilities. The petrochemical industry has been required to implement the Leak Detection and Repair (LDAR) program and cut NMVOC emissions by 30% by 2017. In addition, a wide range of solvent-using activities also increases NMVOC emissions. Solvents appear in many industrial processes, such as dissolving substances, providing media for chemical reactions, and acting as a dispersion medium for coatings. High solids and waterborne paints contain much fewer organic chemicals, and powder coatings and liquid coatings are both solvent-free. The substitution of these new solvents and coating techniques represents the latest requirement of China's emission standards.

6) Eliminate small coal-fired industrial boilers. China shut down all coal boilers with capacities of smaller than 7 MW in urban areas by the end of 2017 and cleaned all existing and new large boilers with $SO_2$ and particulate control technologies. The elimination of small coal-fired boilers in suburban and rural areas is still in progress.

7) Replace residential coal use with electricity and natural gas. Direct coal-burning in the residential sector is being replaced with natural gas and electricity to tackle air pollution in the countryside (China, 2018). China is striving to switch to electricity and gas-powered heating from coal in millions of residences in North China. To facilitate this fuel switch, northern Chinese provinces have cut nonpeak household power prices to reduce the cost of electric heating, and new gas heating systems are being built in suburban and rural regions. These policies can reduce coal use in the residential sector.

8) Strengthen vehicle emissions standards, retire old vehicles, and improve fuel quality. Tightened emission standards have also driven automakers to adopt cleaner technologies. Fuel economy standards have allowed automakers to reduce the amount of fuel use by new cars from 8.0 L 100 km$^{-1}$ in 2010 to 6.9 L 100 km$^{-1}$ in 2015, and the 2020 target is 5.0 L 100 km$^{-1}$ (China State Council, 2016). These fuel economy data are based on laboratory tests under the European standard driving cycle, while the real-world fuel consumption rates are typically 15% higher than these tested values (Huo et al., 2011) because the European test procedure cannot reflect the real urban and highway driving conditions in China. The latest Euro 5/V emission standards were implemented in 2017, and newly registered vehicles must comply with these more stringent emission standards. Additionally, all "yellow label" vehicles were eliminated by the end of 2017. "Yellow label" vehicles refer to gasoline and diesel vehicles that fail to meet Euro 1 and Euro III standards, respectively.

## 3 Methods and Data

### 3.1 Bottom-up emission inventory

Here, we use the framework of the MEIC (Multi-resolution Emission Inventory for China, http://www.meicmodel.org) to estimate China's anthropogenic emissions from 2010–2017. MEIC is a bottom-up emission inventory model which covers 31 provinces in mainland China and includes ~700 anthropogenic sources. Emissions for each source in each province are estimated as follows:

$$Emis_{i,j,k} = A_{i,j} \times \sum_m \left( X_{i,j,m} \times EF_{i,j,k,m} \times \sum_n \left( C_{i,j,m,n} \times \left( 1 - \eta_{k,n} \right) \right) \right) \tag{1}$$

where $i$ represents the province, $j$ represents the emission source, $k$ represents the air pollutants or $CO_2$, $m$ represents the technologies for manufacturing, $n$ represents the technologies for air pollution control, $A$ is the activity rate, $X$ is the fraction of a specific manufacturing technology, $EF$ is the unabated emission factor, $C$ is the penetration of a specific pollution control technology, and $\eta$ is the removal efficiency. The details of the technology-based approach and source classifications can be found in Zhang et al. (2007, 2009), Lei et al. (2011), and Li et al. (2017b).

The underlying data in the MEIC model are gathered from different sources. Activity rates of energy consumptions by fuel type, by sector, and by province are derived from Chinese Energy Statistics (National Bureau of Statistics, 2018a, 2018b; National Energy Administration, 2018). Productions of various industrial products and penetration of different technologies are collected from a wide variety of statistics (for details, please refer to Lu et al., 2010, Lei et al. 2011). We also use

unpublished data from the Ministry of Ecology and Environment to supplement the technology penetration data which are absent in statistics (Qi et al., 2017, Zheng et al., 2017). These data are collected from each plant by local agencies, and then managed and verified by Ministry of Ecology and Environment. The information adopted in this study include pollution control technologies, penetrations, and efficiencies for electric generators, cement factories, iron- and steel-making furnaces, and glass

kilns in each province, which are used to calibrate emission control levels (i.e., $C$ and $\eta$ in Eq. (1)) in the bottom-up inventory. Detailed activity rates by province for the year 2017 are not available when the emission trends presented in this work were developed. In this case, we used national activity data for the year 2017 (National Bureau of Statistics, 2018b; National Energy Administration, 2018) and downscaled these national total data to each province using the weighting factors from 2016 data at the provincial level. Unabated emission factors in MEIC are compiled from a wide range of previous studies, for instance,

$SO_2$ from Lu et al. (2010), $NO_x$ from Zhang et al. (2007), NMVOC from Li et al. (2014), CO from Streets et al. (2006), primary aerosols ($PM_{10}$, $PM_{2.5}$, BC, and OC) from Lei et al. (2011), and $CO_2$ from Liu et al. (2015a). We then override those data by local emission factors summarized in Li et al. (2017c) wherever available, to represent the most recent progress on emission factor developments in China.

Emissions from power plants are estimated following the unit-based approach developed by Liu et al. (2015b). In summary,

we track the emissions of each unit from electricity generation, fuel quality, and the progress in emission control using unit-specific parameters. Emissions from on-road vehicles are estimated using a county-level emission model developed by Zheng et al. (2014), which resolves the spatial-temporal variability of vehicle ownership, fleet turnover (i.e., new technology penetration), and emission factors. Detailed documentation of the method and data for power plants and on-road vehicles can be found in Liu et al. (2015b) and Zheng et al. (2014), respectively.

**3.2 Index Decomposition Analysis**

We use Index Decomposition Analysis (IDA) to study the driving forces of China's anthropogenic emissions from 2010–2017. IDA is one of the major techniques used to analyze the impact of changes in indicators on emission trends (Hoekstra and van den Bergh, 2003). The IDA method is described as follows:

Eq. (1) can be converted to a matrix form using the following formula:

$$Emis_{i,j,k} = A_{i,j}\mathbf{x}\mathbf{E}\boldsymbol{\eta} \tag{2}$$

The technology distribution factors $X_{i,j,m}$ in Eq. (1) are assembled into the row vector $\mathbf{x}$, and the relevant unabated emission factors $EF_{i,j,k,m}$ are assembled into a diagonal matrix $\mathbf{E}$. The column vector $\boldsymbol{\eta}$ represents the average removal efficiencies weighted by the penetration rates $C_{i,j,m,n}$ of all types of pollution control technologies. According to Eq. (2), over a given period of time, any changes in emissions can be decomposed into their component driving factors using Eq. (3).

$$\Delta Emis_{i,j,k} = \Delta A_{i,j}\mathbf{x}\mathbf{E}\boldsymbol{\eta} + A_{i,j}\Delta\mathbf{x}\mathbf{E}\boldsymbol{\eta} + A_{i,j}\mathbf{x}\Delta\mathbf{E}\boldsymbol{\eta} + A_{i,j}\mathbf{x}\mathbf{E}\Delta\boldsymbol{\eta} \tag{3}$$

where $\Delta$ is the difference operator. The four multiplicative terms in Eq. (2) are converted into four additive terms in Eq. (3). Each additive term represents the contribution of one driving factor to the changes in emissions, while all other factors are

kept constant. For example, $\Delta\boldsymbol{\eta}$ is the change in pollutant removal efficiencies and the last term in Eq. (3) represents the change in total emissions caused by end-of-pipe abatement measures, with the activity range $A_{i,j}$, technology distribution $\mathbf{x}$, and unabated emission factor $\mathbf{E}$ assumed to be constant.

Technically, the decomposition of four factors in Eq. (3) has 4! = 24 unique first-order decomposition results. In this study, we use the average of all possible first-order decompositions (Dietzenbacher and Los, 1998) in the analysis of emission drivers. By way of illustration, one of the 24 possible decompositions is shown in Eq. (4).

$$
\begin{aligned}
\Delta Emis_{i,j,k} &= Emis_{i,j,k}(t_1) - Emis_{i,j,k}(t_0) \\
&= \Delta A_{i,j}\mathbf{x}(t_1)\mathbf{E}(t_1)\boldsymbol{\eta}(t_1) + A_{i,j}(t_0)\Delta\mathbf{x}\mathbf{E}(t_1)\boldsymbol{\eta}(t_1) \\
&\quad + A_{i,j}(t_0)\mathbf{x}(t_0)\Delta\mathbf{E}\boldsymbol{\eta}(t_1) + A_{i,j}(t_0)\mathbf{x}(t_0)\mathbf{E}(t_0)\Delta\boldsymbol{\eta}
\end{aligned}
\tag{4}
$$

The decomposition analysis generates a four-dimensional array with dimensions that represent the year (2010–2017), province (size=31), emission source (size>700), and emission drivers (i.e., $A$, $\mathbf{x}$, $\mathbf{E}$, and $\boldsymbol{\eta}$ in Eq. (2)). This means that for each pollutant, the year-to-year change in emissions can be attributed to the drivers of $A$, $\mathbf{x}$, $\mathbf{E}$, and $\boldsymbol{\eta}$ by source and by province. $A$ is the activity effect (e.g., fuel combustion), and the other three factors constitute the overall effect of air pollution control. This study is mainly concerned with source contributions rather than province contributions; hence, we sum the 4-D array of the decomposition analysis results along the province dimension and perform the following analysis at the country scale.

## 3.3 Satellite-based and *in situ* observations

We adopt atmospheric observations to evaluate and validate the emission trends estimated in this study. We use $NO_2$ column retrievals from the DOMINO V2 product (Boersma et al., 2011) and $SO_2$ column retrievals from the OMI V3 product (Krotkov et al., 2015). The 2010–2017 trends of satellite observations are calculated over East China, where anthropogenic sources are dominant relative to natural sources and are compared against the emission trends of $NO_x$ and $SO_2$, respectively. Several recent papers have used satellite retrievals to infer recent trends in emissions from East Asia or China. These results are summarized in this study and compared to our emission estimates. We also collect surface-level $SO_2$, $NO_2$, and $PM_{2.5}$ concentration data from national air quality monitoring stations (http://106.37.208.233:20035/) for the period of 2013–2017. These real-time monitoring stations were established in 2013 and had the ability to report hourly concentrations of criteria pollutants from over 1400 sites in 2017.

## 4 Results and discussion

### 4.1 Emission trends

China's anthropogenic emissions are estimated to have declined by 62% for $SO_2$, 17% for $NO_x$, 27% for CO, 38% for $PM_{10}$, 35% for $PM_{2.5}$, 27% for BC, and 35% for OC since 2010 (Table 1). Most of these emission reductions have been achieved

since 2013 when the Clean Air Action was enacted and implemented. $SO_2$ and $NO_x$ are the only air pollutants that were incorporated into national economic and social development plans with emission reduction targets in China. The 12th Five-Year Plan required the total national emissions of $SO_2$ and $NO_x$ to be cut by 8% and 10% from 2011–2015, respectively, while the actual reductions were much larger than planned due to the more stringent pollution control requirements implemented after 2013. During this period, our estimates suggest that CO emissions decreased by 23%, whereas $CO_2$ emissions were flat, reflecting China's improved combustion efficiency and emission control. The years since 2013 also observed a sharp drop in particulate emissions, in contrast with the flattening emissions observed before 2013. This trend is more evident for coarse particles because they are more easily removed by end-of-pipe abatement measures. Given that China's economy is growing rapidly, China's emissions are decoupling from population, economic, and energy consumption growth (Fig. 2). China's gross domestic product grew by 7.6% per year from 2010 and achieved 67% growth by 2017; however, China's emissions flattened out from 2010–2013, followed by a significant decrease after 2013 according to our calculations. In contrast, NMVOC emissions are estimated to have increased by 11% and $NH_3$ emissions remained flat from 2010–2017; these trends were mainly due to the absence of effective emission control measures.

We present the sectoral trends of China's emissions in Figs. 3 and 4. The most important sector identified by our estimates is the industrial sector, which is the dominant source of $SO_2$, $NO_x$, $PM_{10}$, $PM_{2.5}$, and $CO_2$ emissions during 2010–2017, accounting for average values of 60%, 38%, 57%, 50%, and 53% of total emissions, respectively. The industrial sector is the driver of changes in 2010–2017 emissions for these pollutants except $NO_x$ and $CO_2$, and it also drives down CO and BC emissions. The power sector, though accounting for more than half of burning coal, is not a dominant contributor to the emissions of any pollutant. The reason for this is that upgrading plants with pollution control equipment in the 11th Five-Year Plan (2006–2010) significantly reduced $SO_2$ and particulate emissions from power plants (Liu et al., 2015b), and the remaining part is not compared to industrial emissions. With upgraded emission standards and the spread of the "ultralow emission" technique, the new emission limit values have further driven down power plant emissions, which is the dominant driving force of the decrease in $NO_x$ emissions, while industrial combustion sources lack an effective control on $NO_x$.

The residential sector is the dominant source of CO, BC, and OC, to which it contributes average values of 40%, 49%, and 79% to national emissions, respectively, and is the second-most important source of $PM_{10}$ (31%) and $PM_{2.5}$ (38%); its relative contributions of these components have increased while industrial emissions have considerably decreased. The residential sector drives OC emissions down and contributes to the reductions of CO, NMVOC, and particulate matter. The transportation sector accounts for 17% of CO emissions, 19% of NMVOC emissions, and 31% of $NO_x$ emissions. The increase in fuel consumption drives up transport $CO_2$ emissions, with an increase of 43% from 2010–2017 that is faster than that of any other emission source sector. Solvent use is a major contributor to the increase in NMVOC emissions. Solvent emissions are estimated to have increased by 52% since 2010, making them the largest contributor (36%) to NMVOC emissions in 2017, while the share of this sector was only 27% in 2010. The agricultural sector is the dominant source of $NH_3$ emissions, as it contributes to 93% of total emissions. $NH_3$ emissions have remained constant because agriculture and rural activities showed small interannual variations.

**4.2 Drivers of China's emissions**

The effect of air pollution control can partially or totally offset the additional emissions caused by growing activity rates, and the combination of pollution control and activity growth entirely determines China's emission pathways (Fig. 5). Based on the drivers of emissions, we can classify air pollutants into two categories, namely, activity-driven increasing pollutants and pollution control-driven decreasing pollutants (Fig. 5). NMVOC and $NH_3$ belong to the former category. Their emissions have continued to increase at a constant rate from 2010–2017, primarily driven by activity growth. Assuming that activity rates are frozen at their 2010 levels (Fig. 3), the NMVOC emissions could decrease by 21% from 2010 to 2017 due to the emission controls on residential and transport sectors. These emission reductions were far outweighed by the growing use of solvent for paints, coatings, and chemical industry, which consequently drove up their total emissions. The solvent used for paints increased by 110% from 2010, which is attributed to the increasing demand to coat buildings, cars, and machinery due to the rapid increase in the area of the newly built house (+52%) and the production of vehicles (+54%). The solvent used in chemical industry also rose at a fast rate due to the increase in industrial production (e.g., ethylene production grew by 28% since 2010), which makes them the second largest contributor to NMVOC growth after paints. For $NH_3$, the lack of control measures has caused its emissions to correlate well with activity; thus, its emissions do not decline similar to the regulated pollutants that have experienced progressive emission control.

The other air pollutants followed distinct emission pathways before and after 2013 (Fig. 5). According to our estimates the emissions of these pollutants slightly increased (e.g., $SO_2$ and $NO_x$) or remained flat (e.g., CO and particulate matter) during 2010–2013 because emission mitigation just counterbalanced the additional emissions caused by growing activities. China's fuel combustion increased by 15.2% from 2010–2013 (Fig. 6), and its industrial production increased by 14–35% in different industries. During this period, China's clean air actions mainly focused on upgrading emission standards for the power and industrial sectors. These measures effectively offset the growth in activities but were not stringent enough to reverse the growing trends in emissions; therefore, air pollutant emissions remained stable from 2010–2013.

After 2013, emissions of all air pollutants except NMVOC and $NH_3$ are found to have reduced as a result of pollution controls. China's fuel combustion and industrial production have flattened out since 2013 (Fig. 6), while high-efficiency mitigation measures have been increasingly implemented in all emission source sectors, as required by the Clean Air Action. Scenario analysis suggests that the effect of pollution control rapidly removes air pollutants and consequently drives down China's emissions (Fig. 3). Assuming that pollution control is frozen at 2010 levels, $SO_2$ emissions in 2017 could increase by 167% compared to the actual data, $NO_x$ and TSP emissions could increase by 38% and 111%, respectively, and other pollutants could see increases of 23–66%. The different reduction rates of air pollutant emissions are determined by the source sector distributions and emission mitigation efforts of each sector. For example, this decrease is most notable for $SO_2$ (emissions are estimated to have decreased by 59% from 2013–2017) because the dominant source sectors (i.e., power and industry) both significantly reduced their emissions. The decrease in emissions is smallest for $NO_x$ (21% of emissions cut from 2013–2017 based on the analysis) because the power sector was the major contributor to emission reduction but only accounted for one-

third of total emissions. To understand the underlying drivers of emission reduction, we decompose the avoided emissions due to pollution control (i.e., the sum of contributions from $\mathbf{x}$, $\mathbf{E}$, and $\boldsymbol{\eta}$) into sectors (Fig. 7) to identify the main drivers underlying key source categories. We select the year of 2017, which exhibited the largest reduction in emissions according to our calculations, to perform this analysis.

1) The power sector. The generation of electricity from hydrocarbon fuels in China has increased by 33% since 2010, which has led to increases of 1.2 Tg $SO_2$ and 1.7 Tg $NO_x$ in 2017 compared with their levels in 2010 (Fig. 7). Mitigation efforts have yielded reductions of 7.1 Tg $SO_2$ and 6.1 Tg $NO_x$ and thus totally offset the emissions caused by growing activities. The reduction of emissions was achieved through the "ultralow emission" standard. To fulfill the stringent standards, flue gas desulfurization (FGD) and selective catalytic reduction (SCR) systems have been increasingly installed at utilities in coal-fired

power plants, with penetration rates reaching >95% in 2017. Of the current power plants, 71% have operated close to the design performance of "ultralow emission" levels (China, 2018).

2) The industrial sector. Mitigation measures have yielded reductions of 9.5 Tg $SO_2$, 0.9 Tg $NO_x$, 38.1 Tg CO, 3.4 Tg $PM_{2.5}$, 0.3 Tg BC, and 0.3 Tg OC from the industrial sector in 2017 compared with their levels in 2010 (Fig. 7). For $SO_2$, shutting small industrial boilers and cleaning larger ones have contributed the most to emission reductions. In particular, small coal

boilers (≤7 MW) located in urban areas were eliminated by the end of 2017, and large boilers have extensively used sorbent injection technologies to remove $SO_2$ from exhaust gases. For other pollutants, the most effective measures include strengthening industrial emission standards, eliminating outdated industrial capacity, and phasing out small, high emitting factories. The improvements in combustion efficiency and oxygen blast furnace gas recycling are the largest drivers of declining CO emissions, and the wide use of high-efficiency dust collectors (e.g., electrostatic precipitators and fabric filters)

in manufacturing industries has successfully removed particulate matter. In addition, the desulfurization of sinter plant gases accounts for 8% of $SO_2$ emission reductions, and denitrification in cement kilns accounts for 6% of $NO_x$ emission reductions. The low-sulfur, low-ash coals resulting from fuel quality improvements have also helped reduce $SO_2$ and particulate emissions.

3) The residential sector. The emission reductions achieved by the residential sector are primarily driven by the decrease in activities mainly caused by replacing coal with natural gas and electricity (Fig. 7), which yielded reductions of 13.5 Tg CO,

1.0 Tg $PM_{2.5}$, 0.1 Tg BC, and 0.7 Tg OC in 2017 compared with their 2010 levels. Additionally, pollution controls caused additional reductions of 0.5 Tg CO, 0.3 Tg $PM_{2.5}$, 0.1 Tg BC, 0.1 Tg OC, and 1.0 Tg $SO_2$. The decreases in activity rates reflect both long-term changes in fuel mixtures, i.e., from traditional biofuels to commercial energy, and short-term measures to replace coal with clean energy. Pollution control policies have promoted the use of clean stoves and the switch from raw coal to clean coal briquettes with lower levels of sulfur and ash.

4) The transportation sector. Pollution controls on the transportation sector have exactly counterbalanced the growing emissions due to vehicle growth (Fig. 7). China's vehicle ownership reached 209 million in 2017; this value is 2.7 times larger than its 2010 value. Growing activities yielded increases of 22.2 Tg CO, 3.6 Tg NMVOC, and 1.4 Tg $NO_x$ in 2017 compared with their 2010 levels, while pollution control measures have yielded reductions of 29.0 Tg CO, 4.8 Tg NMVOC, and 1.3 Tg $NO_x$. The reduction of emissions is mainly achieved through fleet turnover, which means that old vehicles are being replaced

by newer, cleaner models subjected to tougher emission standards. China has scrapped all the old vehicles that don't meet the more stringent emission standards, i.e., "yellow label" vehicles, by the end of 2017. The number of vehicles scrapped in each province is recorded by the local government, and these scrapped vehicles are banned from roads and sent to wrecking yard for recycling. Consequently, the estimated share of fuel consumption by Euro 4 and Euro 5 vehicles increased from 2% in 2010 to 66% in 2017. The effects of these changes on reducing particulate emissions are smaller because transport contributes only a small fraction of total particulate emissions. For NMVOC, the transport sector is the only sector that has seen a deep cut in emissions. More than 80% of NMVOC emission reductions are achieved from tailpipe exhaust sources, which have caused evaporative emissions to be the primary source (>60%) of the remaining NMVOC emissions from transport.

## 4.3 Comparison with observations and implication for uncertainties

Comparison of trends in $PM_{2.5}$ precursor emissions with satellite and ground-based $PM_{2.5}$ concentrations are presented in Fig. 8a. All data are normalized to the year 2013 because it is the only year that all data are available. Satellite-derived $PM_{2.5}$ concentrations presented a relatively flat trend during 2010 and 2013 (e.g., Fontes et al., 2017; Liang et al., 2018; Lin et al., 2018), corresponding to small variations in emissions of different precursors estimated for the same period. Satellite-based $PM_{2.5}$ concentrations over China decreased by 18% from 2013–2015 (Lin et al., 2018), in good agreement with the trend in surface $PM_{2.5}$ concentrations over 74 cities. During 2013–2017, surface $PM_{2.5}$ concentrations over 74 cites decreased by 35%, while emissions of $PM_{2.5}$ precursors over China presented various changing rates. We estimate a faster decrease in $SO_2$ emissions than the observed surface $PM_{2.5}$ concentrations, while the estimated decrease rates of $NO_x$ and $NH_3$ emissions were slower than the observed $PM_{2.5}$ concentrations. This phenomenon was qualitatively confirmed by an observed large decrease of sulfate and increased relative contribution of nitrate in $PM_{2.5}$ compositions from 2013–2017 (Shao et al. 2018). However, the quantitative relationship between $PM_{2.5}$ concentrations and precursor emissions will require further studies with chemical transport modelling.

Trends in $NO_x$ and $SO_2$ emissions are generally consistent with satellite- and ground-based observations during 2010–2017 (Fig. 8b and Table 2). Specifically, rapid decreases in $SO_2$ and $NO_x$ emissions after 2013 are confirmed by satellite-based observations. We estimate that $NO_x$ and $SO_2$ emissions in Eastern China decreased by 21% and 59% during 2013–2017 respectively, lower than 30% and 73% decrease in OMI observed $NO_2$ and $SO_2$ columns for the same region and time period. Surface $SO_2$ concentrations decreased by 57% over Eastern China for the period of 2013–2017, in good agreement with the estimated emission trend. In contrast, surface $NO_2$ concentrations only decreased by 9% for the same time, significantly lower than the estimated trend in $NO_x$ emissions.

Different trends between emissions and concentrations could be attributed to many factors. First, temporal and spatial patterns of emissions and concentrations are impacted by variations in meteorology, atmospheric transport, and chemical reactions. Interannual variabilities can result in remarkable variations in column and surface concentrations (Uno et al., 2007), which may partly explain the disagreement between changes in emissions and observations for a signal year (e.g., the year 2011 in Fig. 8b). Surface observations are more sensitive to surface emissions than high-stack emissions and satellite-based column

observations are more visible to high-stack emissions due to different transport patterns, in which both will contribute to the differences when comparing trends. In addition, chemical partitioning of $NO_2/NO_x$ may also contribute to discrepancies between trends in emissions and observations (Lamsal et al., 2011, Valin et al., 2011). Taking the above factors into account, inverse modeling (IM) approaches were developed to derive top-down emissions constrained by observations. Table 2

presented recently published estimates on top-down emission trends over China using IM approaches. As shown in Table 2, the discrepancies between bottom-up and top-down emission estimates are not always narrowed, indicating uncertainties from other aspects might exist.

Second, uncertainties in observations can also contribute to the discrepancies. *In-situ* observations are usually thought to be more accurate. However, surface $NO_2$ concentrations obtained from national monitoring network relied on chemiluminescence

measurements, which can significantly overestimate $NO_2$ concentrations (Lamsal et al., 2010) and then contribute to discrepancies between emissions and surface observations. Satellite retrievals are subject to larger uncertainties, for instance, tropospheric $SO_2$ columns are quite uncertain due to difficulties in isolating anthropogenic $SO_2$ signals from ozone and volcanic $SO_2$ (Krotkov et al., 2006). Most uncertainties in satellite retrievals are systematic and cancelled when trends are compared. However, influences of aerosols on satellite $NO_2$ retrievals (Lin et al., 2015) may impact the reliability of the $NO_2$ column

trend due to the large decrease in aerosol concentrations over the discussed period. $SO_2$ columns are less sensitive to small and near surface emissions (Li et al., 2017a), which may lead to an underestimation of the $SO_2$ budget in China for most recent years and a disagreement between emission and $SO_2$ column trend when high-stack emissions (e.g., power plants) were significantly reduced.

Last but not least, emissions estimates are uncertain due to incomplete knowledge of underlying data (Zhao et al., 2011; Li et

al., 2017c). Similar magnitudes of uncertainties are expected comparing to our previous work (e.g., Zhang et al., 2009; Lei et al., 2011; Lu et al., 2011; Li et al., 2017c) since similar methodologies and data sources are used. In general, uncertainties are smaller for species which emissions are dominant by large sources (e.g., $SO_2$ and $NO_x$) but larger for species which emissions are mainly contributed by scattered emitting sources (e.g., BC and OC). Many of the uncertainties in bottom-up emissions are also systematic and may have less impact on emission trends (Lu et al., 2011), but non-compliance with regulations due to

lack of inspection will lead to differences between estimated and real-world efficiencies of emission control facilities (e.g., Wang et al., 2015) and impact the validity of estimated emission trend. Specifically, the effectiveness of the measures targeting the small and scattered emitting sources (e.g., phase out small, high emitting factories and eliminate small coal-fired industrial boilers) are difficult to validate, which may lead to higher uncertainty ranges in emission estimates for most recent years.

**5 Concluding Remarks**

From 2010–2017, China reduced its anthropogenic emissions by 62% for $SO_2$, 17% for $NO_x$, 27% for CO, 38% for $PM_{10}$, 35% for $PM_{2.5}$, 27% for BC, and 35% for OC according to our estimates. Compared to observations, the trends in $NO_x$ and $SO_2$ emissions are broadly consistent with OMI satellite and ground-based measurement, and the emissions trends of $PM_{2.5}$

precursors agree well with changes in PM$_{2.5}$ compositions over China. Some differences between emissions trends and observations could be attributed to uncertainties in atmospheric measurement and emissions estimates, as well as the mismatch in their spatial-temporal patterns due to chemical processes in the atmosphere. Most of the emission reductions were achieved after 2013, and the Index Decomposition Analysis confirms that emission control measures have been the dominant driver of this declining emission trend. Pollution controls on the power and industrial sectors are the most effective measures, which have contributed 56–94% of total avoided emissions due to stringent mitigation policies, such as strengthening emission standards, eliminating outdated industrial capacity, and phasing out small, high emitting factories. Emissions from transport tend to remain flat because the effect of air pollution control is offset by the additional emissions from growing activities. The residential sector has reduced its emissions mainly through the substitution of clean fuels. From 2010–2017, NMVOC emissions are estimated to have increased by 11% and NH$_3$ emissions flattened because China lacked effective emission control measures on NMVOC and NH$_3$ in current policies.

All these emissions reductions from 2010–2017 were driven by the objective to reduce PM$_{2.5}$ pollutions in China. For years after 2017, all cities that exceed the 35 μg m$^{-3}$ annual standard are further required to reduce annual average PM$_{2.5}$ concentrations by 18% in 2020 compared to their 2015 levels. Since the annual average limit of PM$_{2.5}$ is exceeded in many Chinese cities currently, the 2020 air quality target will continue driving down China's air pollutant emissions in the future. With emissions going down, the contributions of once-dominant source sectors have decreased, and emissions from other sources have gradually occupied larger proportions (Fig. 9). The change in the sectoral distribution of emissions indicates that it is paramount to shift policy focus to enable further emission reductions. China's clean air policies during 2013–2017 had limited effects on reducing emissions from the residential, off-road, vehicle evaporative, solvent use, and agricultural sectors; therefore, these sectors have significantly increased their contributions from 2010 to 2017 based on our analysis (Fig. 9). The residential sector is estimated to account for 23–50% of SO$_2$ and particulate emissions in 2017, comparable to or even larger than the emissions from the power and industrial sectors. The contribution of off-road transport to NO$_x$ emissions is estimated to have increased from 8% to 12% (Fig. 9b) and thus ranked as the fourth largest single sector in 2017. For road transport, evaporation emissions of NMVOC are larger than tailpipe emissions now because the tailpipe emissions reduced significantly from 2010–2017. A wide range of solvent-using activities drove up NMVOC emissions, therefore solvent use ranked as the largest source sector (Fig. 9d). The agricultural sector currently lacks targets, policies, and measures to control NH$_3$ emissions. These less-controlled emission sources have large potential effects on China's 2020 air quality target, thus China needs to increase its focus on these sources from now on.

**Data availability**

The emission inventory data developed by this work is publically available from https://doi.org/10.6084/m9.figshare.c.4217624.v1, and http://www.meicmodel.org. A supplementary data file containing all underlying data for the figures of the manuscript is also provided in supplementary information. For the data used in this work,

statements have been included about which data are publically available (accessed through references and links). The confidential information used in this study is a firm-level database for key industries in China, which are owned and managed by the Ministry of Ecology and Environment and not available to the public. The firm-level data are used to derive the penetration rates of different emission control technologies ($C$ and $\eta$ in Eq. 1). The role of these confidential data in the estimates of emissions has been clarified in the main text.

## Acknowledgements

This work was supported by the National Key R&D program (2016YFC0201506), the National Natural Science Foundation of China (41625020, 91744310, and 41571130035), and the public welfare program of China's Ministry of Environmental Protection (201509004).

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

| Source sector | Emission source | 2010 | 2011 | 2012 | 2013 | 2014 | 2015 | 2016 | 2017 |
|---|---|---|---|---|---|---|---|---|---|
| Power | Thermal power plants | GB 13223-2003 | GB 13223-2003 | GB 13223-2011 | GB 13223-2011 | GB 13223-2011 | GB 13223-2011 | "ultra-low" emission standard | "ultra-low" emission standard |
| Industry | Flat glass | GB 9078-1996 | GB 26453-2011 | GB 26453-2011 | GB 26453-2011 | GB 26453-2011 | GB 26453-2011 | GB 26453-2011 | GB 26453-2011 |
| | Sinter | GB 9078-1996 | GB 9078-1996 | GB 28662-2012 | GB 28662-2012 | GB 28662-2012 | GB 28662-2012 | GB 28662-2012 | GB 28662-2012 |
| | Coking | GB 16171-1996 | GB 16171-1996 | GB 16171-2012 | GB 16171-2012 | GB 16171-2012 | GB 16171-2012 | GB 16171-2012 | GB 16171-2012 |
| | Iron | GB 9078-1996 | GB 9078-1996 | GB 28663-2012 | GB 28663-2012 | GB 28663-2012 | GB 28663-2012 | GB 28663-2012 | GB 28663-2012 |
| | Steel making | GB 9078-1996 | GB 9078-1996 | GB 28664-2012 | GB 28664-2012 | GB 28664-2012 | GB 28664-2012 | GB 28664-2012 | GB 28664-2012 |
| | Steel rolling | GB 9078-1996 | GB 9078-1996 | GB 28665-2012 | GB 28665-2012 | GB 28665-2012 | GB 28665-2012 | GB 28665-2012 | GB 28665-2012 |
| | Electronic glass | GB 9078-1996 | GB 9078-1996 | GB 9078-1996 | GB 29495-2013 | GB 29495-2013 | GB 29495-2013 | GB 29495-2013 | GB 29495-2013 |
| | Brick | GB 9078-1996 | GB 9078-1996 | GB 9078-1996 | GB 9078-1996 | GB 29620-2013 | GB 29620-2013 | GB 29620-2013 | GB 29620-2013 |
| | Cement | GB 4915-2004 | GB 4915-2004 | GB 4915-2004 | GB 4915-2004 | GB 4915-2013 | GB 4915-2013 | GB 4915-2013 | GB 4915-2013 |
| | Industrial boiler | GB 13271-2001 | GB 13271-2001 | GB 13271-2001 | GB 13271-2001 | GB 13271-2014; Eliminate small coal-fired boilers. | GB 13271-2014; Eliminate small coal-fired boilers. | GB 13271-2014; Eliminate small coal-fired boilers. | GB 13271-2014; Eliminate small coal-fired boilers. |
| | All | / | / | / | / | Phase out outdated industrial capacity; Strengthen emissions standards; Phase out small, high emitting factories; Install VOC emission control facilities | Phase out outdated industrial capacity; Strengthen emissions standards; Phase out small, high emitting factories; Install VOC emission control facilities | Phase out outdated industrial capacity; Strengthen emissions standards; Phase out small, high emitting factories; Install VOC emission control facilities | Phase out outdated industrial capacity; Strengthen emissions standards; Phase out small, high emitting factories; Install VOC emission control facilities |
| Residential | All | No specific regulations | No specific regulations | No specific regulations | Replace coal with electricity and natural gas | Replace coal with electricity and natural gas | Replace coal with electricity and natural gas | Replace coal with electricity and natural gas | Replace coal with electricity and natural gas |
| Transportation | Light duty gasoline vehicle | Euro 3 | Euro 4 | Euro 4 | Euro 4 | Euro 4 | Euro 4 | Euro 4 | Euro 5 |
| | Heavy duty gasoline vehicle | Euro 3 | Euro 3 | Euro 3 | Euro 4 | Euro 4 | Euro 4 | Euro 4 | Euro 4 |
| | Diesel vehicle | Euro III | Euro III | Euro III | Euro III | Euro IV | Euro IV | Euro IV | Euro V |
| | All | / | / | / | / | Strengthen emissions standards; Retire old vehicles; Improve fuel quality | Strengthen emissions standards; Retire old vehicles; Improve fuel quality | Strengthen emissions standards; Retire old vehicles; Improve fuel quality | Strengthen emissions standards; Retire old vehicles; Improve fuel quality |

**Figure 1. China's clean air policies implemented from 2010–2017.**

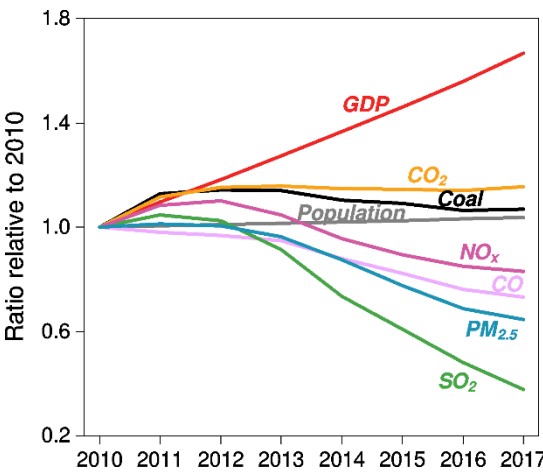

**Figure 2. Emission trends and underlying social and economic factors.** The coal usage is achieved from Chinese Energy Statistics (National Bureau of Statistics, 2018a, 2018b). The GDP and population data come from the National Bureau of Statistics (2018b, 2018c). Data are normalized by dividing the value of each year by their corresponding value in 2010.

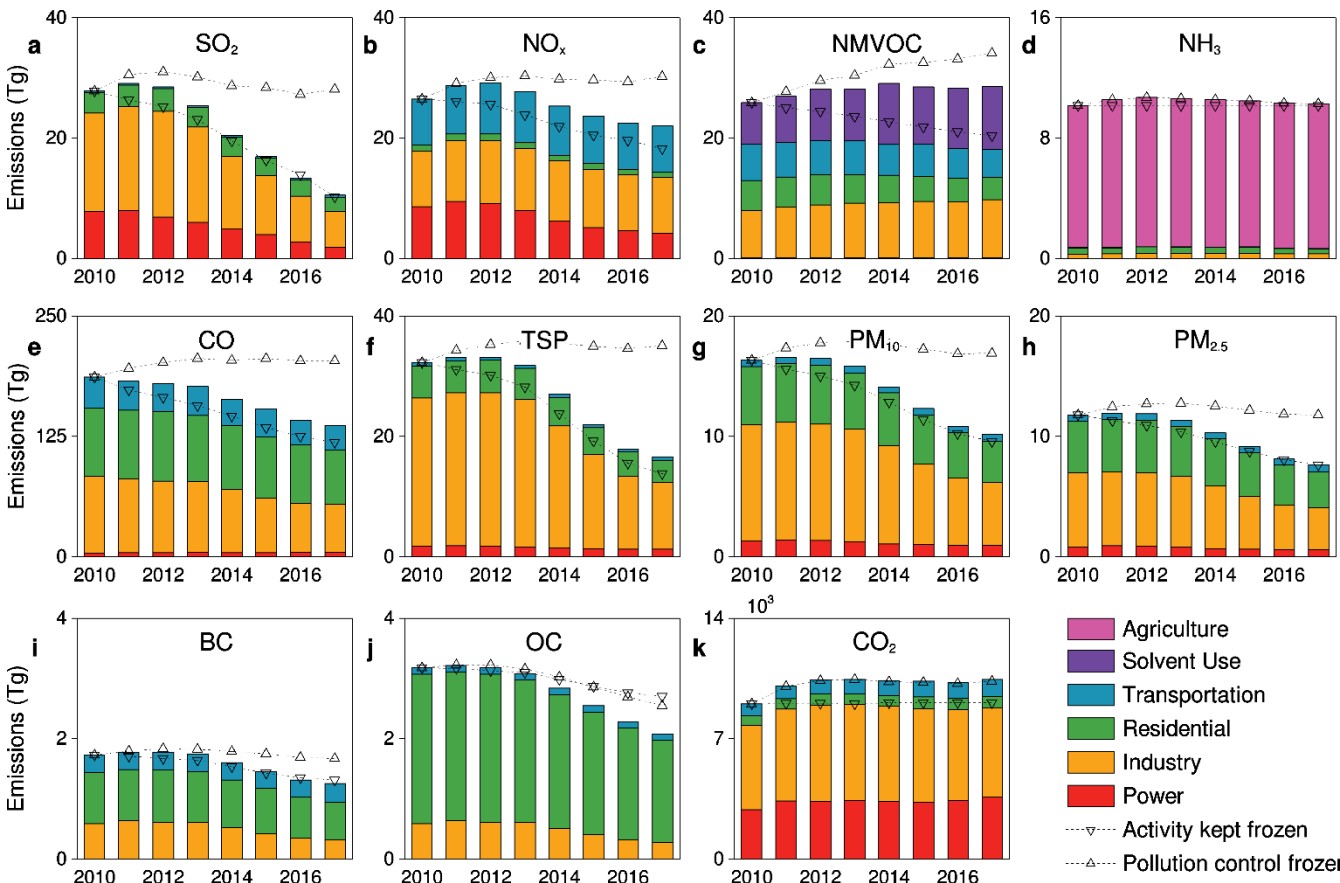

**Figure 3. China's anthropogenic emissions by sector and year.** The species plotted here include (a) $SO_2$, (b) $NO_x$, (c) NMVOC, (d) $NH_3$, (e) CO, (f) TSP, (g) $PM_{10}$, (h) $PM_{2.5}$, (i) BC, (j) OC, and (k) $CO_2$. China emissions are divided into six source sectors (stacked column chart): power, industry, residential, transportation, agriculture, and solvent use. Besides the actual emissions data, two emission scenarios are presented to provide emission trajectories when assuming activity (inverted triangle) or pollution control (upright triangle) frozen at 2010 levels.

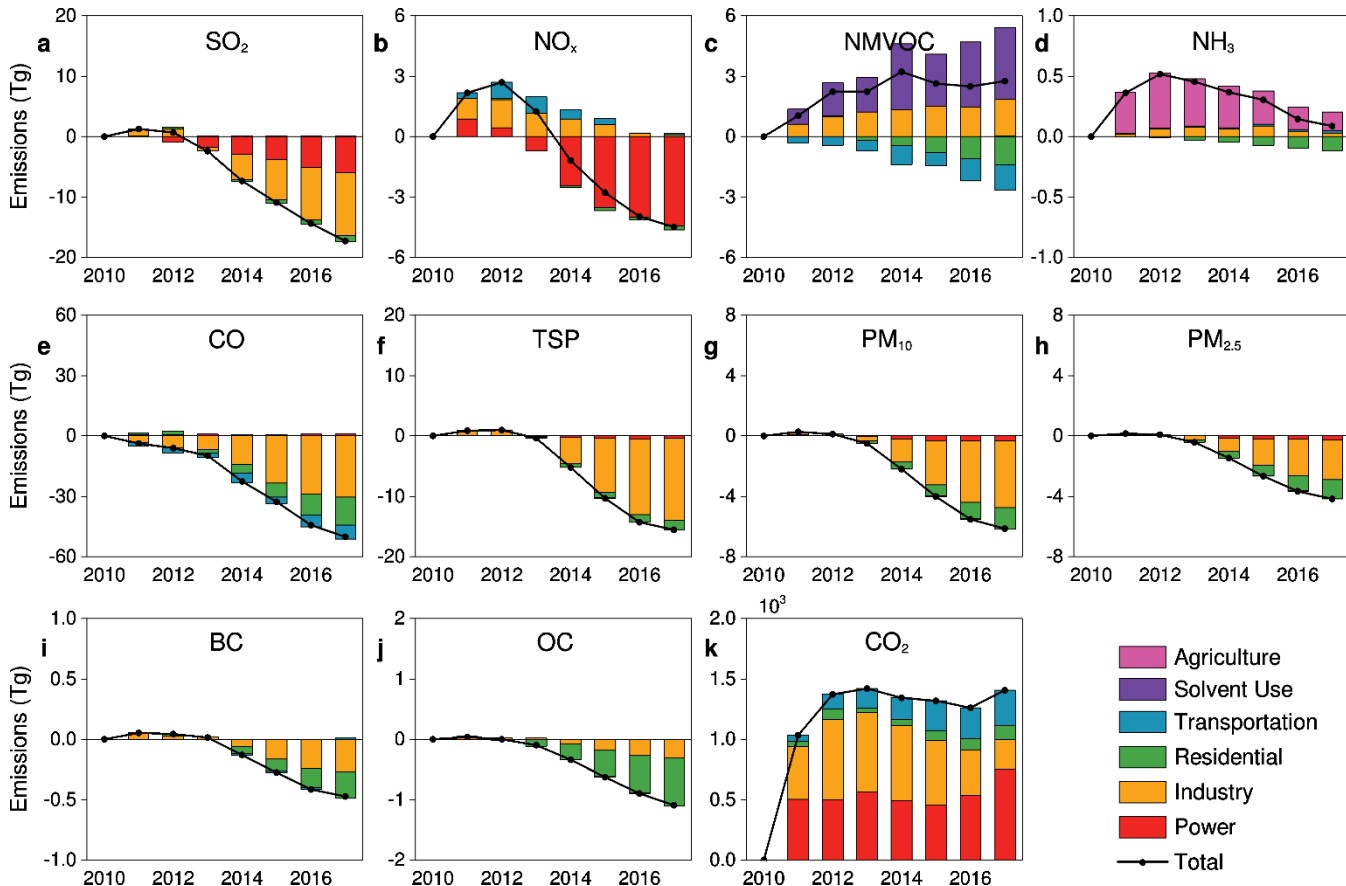

**Figure 4. Changes in China's emissions by sector and year.** The species plotted here include (a) $SO_2$, (b) $NO_x$, (c) NMVOC, (d) $NH_3$, (e) CO, (f) TSP, (g) $PM_{10}$, (h) $PM_{2.5}$, (i) BC, (j) OC, and (k) $CO_2$. The 2010 emissions are subtracted from the emission data for each year to represent the additional emissions compared to 2010 levels. The emission changes are shown by sector (stacked column chart) and as national totals (black curve).

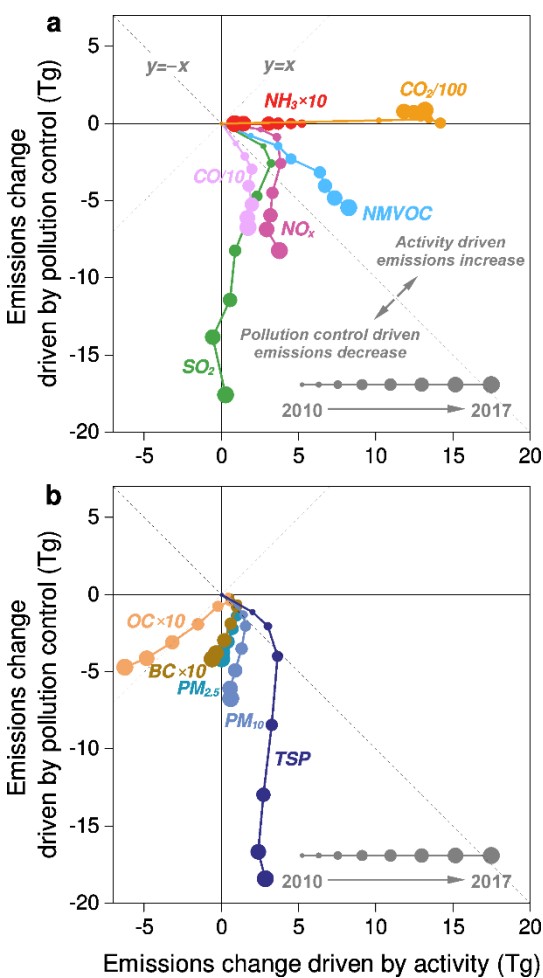

**Figure 5. China's emission pathways from 2010–2017.** Gaseous pollutants are plotted in (a), and particles are plotted in (b). For each pollutant, the years (circle) are plotted according to the emission changes caused by activity (*A* in Eq. (2), x-axis) and pollution control (the sum of **x**, **E,** and **η** in Eq. (2), y-axis). Please refer to Fig. S1 for decomposition analysis results of *A*, **x**, **E,** and **η**. The intersecting lines y=x and y=−x divide the coordinate plane into four sections. Any point in the section on the right side of the two lines reflects increasing emissions due to activity growth, and the points in the section below the two lines reflect decreasing emissions driven by pollution control.

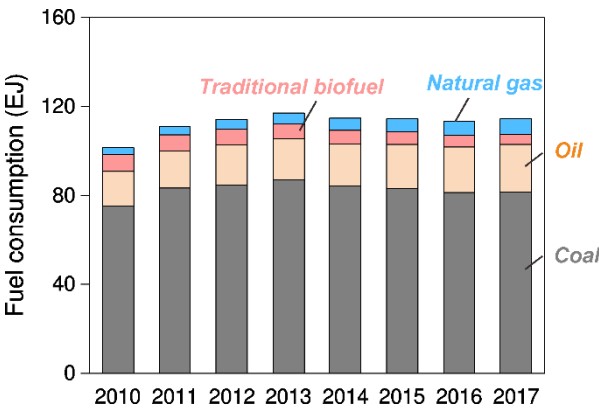

**Figure 6. Energy consumption of hydrocarbon fuels from 2010 to 2017.** Coal includes all coal-based fuels, and oil includes all oil-based fuels. Traditional biofuel includes crop residual and wood.

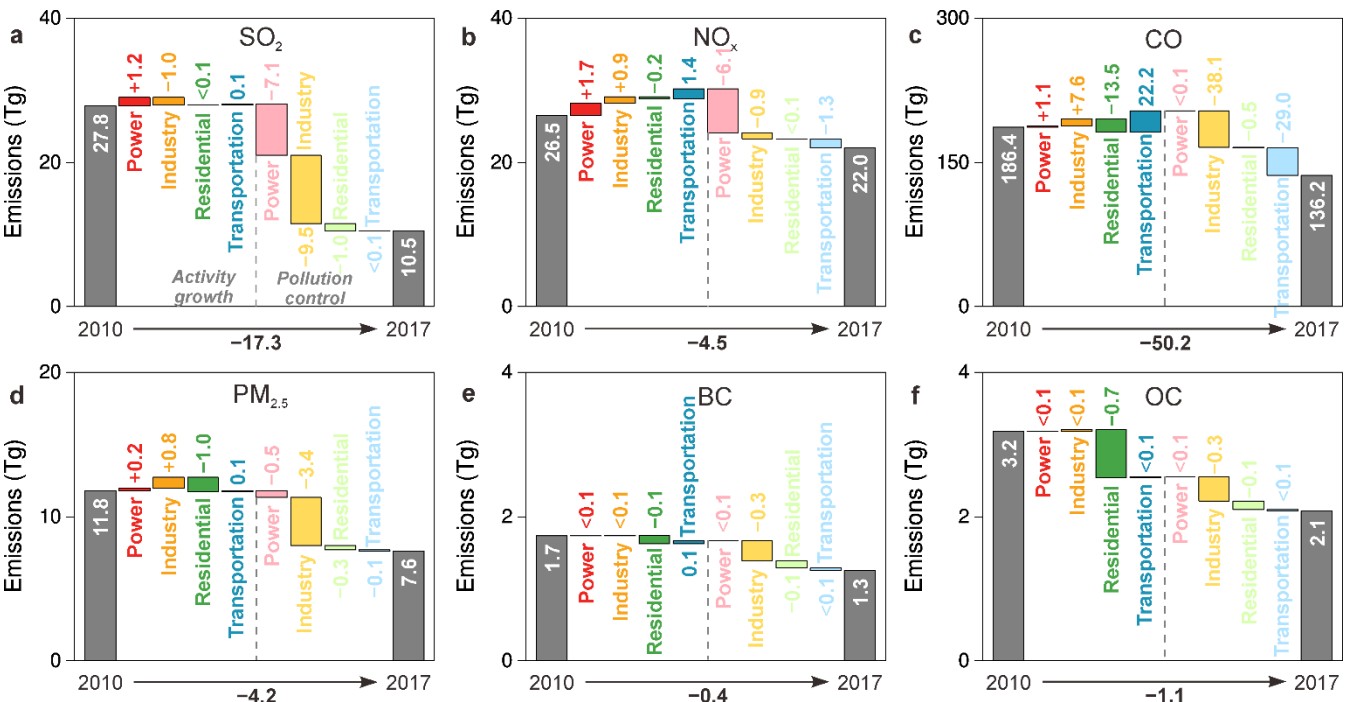

**Figure 7. Drivers of emission changes for different emission species.** The species plotted here include (a) $SO_2$, (b) $NO_x$, (c) CO, (d) $PM_{2.5}$, (e) BC, and (f) OC. For each pollutant, the changes in emissions from 2010 to 2017 (bar) are decomposed into drivers of activity growth (*A* in Eq. (2)) and pollution control (the sum of **x**, **E,** and **η** in Eq. (2)) by source sector.

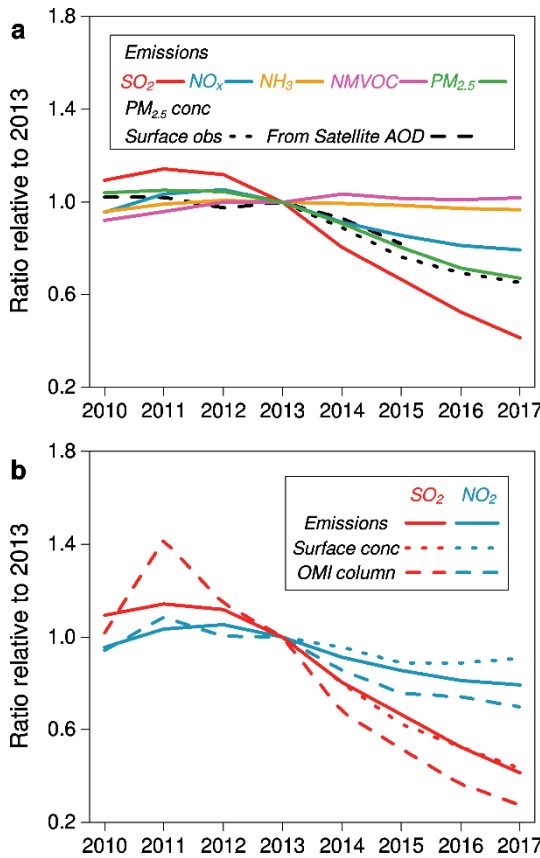

**Figure 8. Emission trends compared with satellite- and ground-based observations.** The satellite-retrieved PM$_{2.5}$ concentrations (black, the dashed curve in a) (Lin et al., 2018) are compared with emission trends of PM$_{2.5}$ precursors in a. The 2010–2017 trends in SO$_2$ (red, the solid curve in b) and NO$_x$ (blue, the solid curve in b) emissions are compared with OMI SO$_2$ (red, the dashed curve in b) and NO$_2$ (blue, the dashed curve in b) tropospheric columns for Eastern China, respectively. Eastern China here includes the provinces of Beijing, Tianjin, Hebei, Shanxi, Shaanxi, Shandong, Henan, Hubei, Anhui, Jiangsu, Shanghai, and Zhejiang. Besides, the 2013–2017 trends in ground-based observations of SO$_2$ (red, the dotted curve in b), NO$_2$ (blue, the dotted curve in b), and PM$_{2.5}$ (black, the dotted curve in a) are also presented. Data are normalized by dividing the value of each year by their corresponding value in 2013.

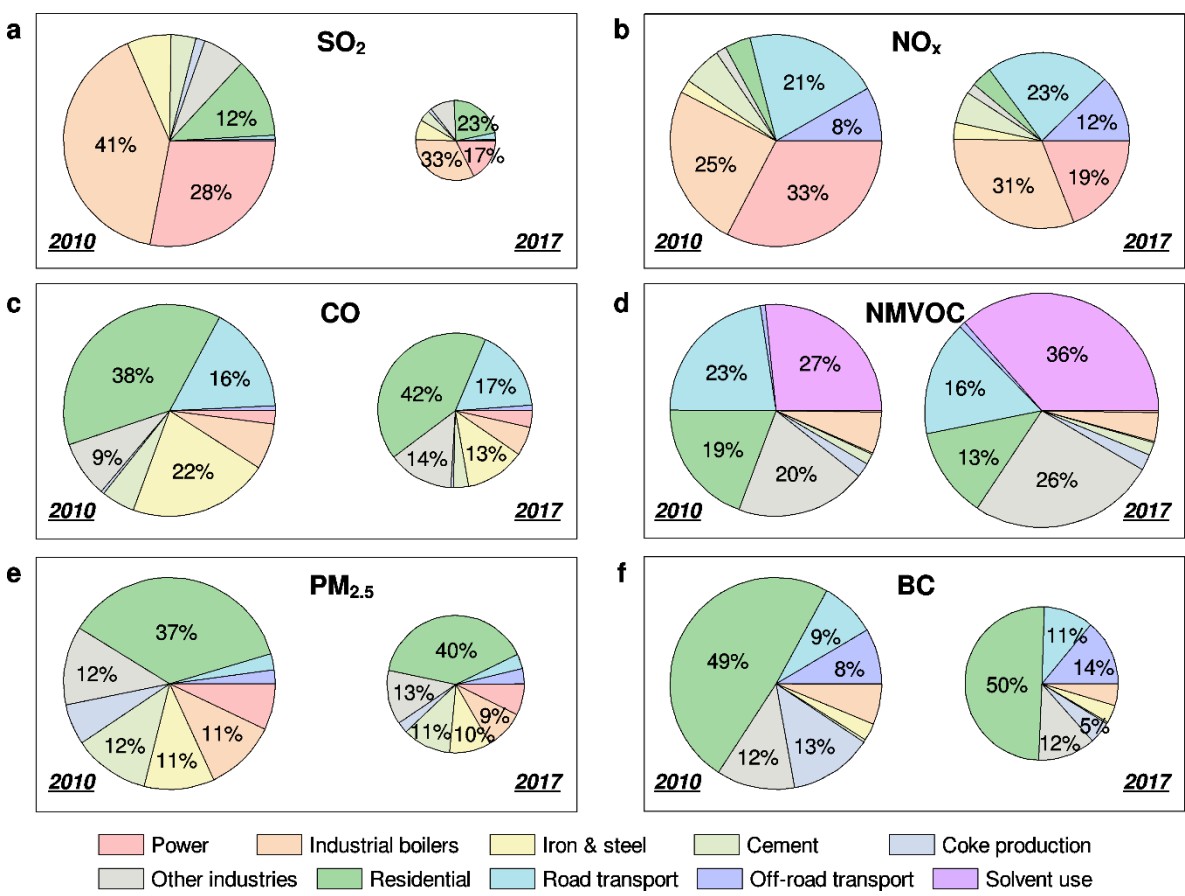

**Figure 9: Changes in emission percentages across source sectors from 2010 to 2017.** The species plotted here include (a) SO$_2$, (b) NO$_x$, (c) CO, (d) NMVOC, (e) PM$_{2.5}$, and (f) BC. For each pollutant, the relative change in the radius of the pie chart from 2010 to 2017 is proportional to the change in emissions.

**Table 1: Anthropogenic emissions of air pollutants and CO₂ in China from 2010 to 2017.**

| Year | $SO_2$[a] | $NO_x$ | NMVOC | $NH_3$ | CO | TSP[b] | $PM_{10}$ | $PM_{2.5}$ | BC | OC | $CO_2$[c] |
|---|---|---|---|---|---|---|---|---|---|---|---|
| Power | 7.8 | 8.6 | 0.1 | 0.0 | 3.8 | 1.7 | 1.3 | 0.8 | 0.0 | 0.0 | 2864.8 |
| Industry | 16.4 | 9.1 | 7.9 | 0.3 | 79.7 | 24.7 | 9.6 | 6.1 | 0.6 | 0.6 | 4914.3 |
| Residential | 3.4 | 1.0 | 5.0 | 0.4 | 70.9 | 5.3 | 4.8 | 4.3 | 0.8 | 2.5 | 567.8 |
| Transportation | 0.2 | 7.7 | 6.1 | 0.0 | 32.0 | 0.6 | 0.5 | 0.5 | 0.3 | 0.1 | 682.5 |
| Agriculture | 0.0 | 0.0 | 0.0 | 9.5 | 0.0 | 0.0 | 0.0 | 0.0 | 0.0 | 0.0 | 0.0 |
| Solvent use | 0.0 | 0.0 | 6.9 | 0.0 | 0.0 | 0.0 | 0.0 | 0.0 | 0.0 | 0.0 | 0.0 |
| 2010 | 27.8 | 26.5 | 25.9 | 10.2 | 186.4 | 32.2 | 16.3 | 11.8 | 1.7 | 3.2 | 9029.4 |
| Power | 7.9 | 9.5 | 0.1 | 0.0 | 4.4 | 1.8 | 1.4 | 0.9 | 0.0 | 0.0 | 3365.2 |
| Industry | 17.3 | 10.2 | 8.5 | 0.3 | 76.3 | 25.4 | 9.8 | 6.2 | 0.6 | 0.6 | 5354.7 |
| Residential | 3.6 | 1.1 | 5.0 | 0.4 | 71.6 | 5.3 | 4.9 | 4.3 | 0.9 | 2.5 | 609.9 |
| Transportation | 0.3 | 8.0 | 5.8 | 0.0 | 30.2 | 0.5 | 0.5 | 0.5 | 0.3 | 0.1 | 735.7 |
| Agriculture | 0.0 | 0.0 | 0.0 | 9.8 | 0.0 | 0.0 | 0.0 | 0.0 | 0.0 | 0.0 | 0.0 |
| Solvent use | 0.0 | 0.0 | 7.6 | 0.0 | 0.0 | 0.0 | 0.0 | 0.0 | 0.0 | 0.0 | 0.0 |
| 2011 | 29.1 | 28.7 | 26.9 | 10.5 | 182.7 | 33.1 | 16.6 | 11.9 | 1.8 | 3.2 | 10065.5 |
| Power | 6.9 | 9.1 | 0.1 | 0.0 | 4.5 | 1.7 | 1.3 | 0.9 | 0.0 | 0.0 | 3361.3 |
| Industry | 17.6 | 10.5 | 8.9 | 0.3 | 74.0 | 25.5 | 9.7 | 6.1 | 0.6 | 0.6 | 5584.5 |
| Residential | 3.7 | 1.1 | 5.0 | 0.4 | 72.4 | 5.4 | 4.9 | 4.4 | 0.9 | 2.5 | 652.1 |
| Transportation | 0.3 | 8.5 | 5.6 | 0.0 | 29.4 | 0.6 | 0.5 | 0.5 | 0.3 | 0.1 | 802.2 |
| Agriculture | 0.0 | 0.0 | 0.0 | 9.9 | 0.0 | 0.0 | 0.0 | 0.0 | 0.0 | 0.0 | 0.0 |
| Solvent use | 0.0 | 0.0 | 8.5 | 0.0 | 0.0 | 0.0 | 0.0 | 0.0 | 0.0 | 0.0 | 0.0 |
| 2012 | 28.5 | 29.2 | 28.1 | 10.7 | 180.2 | 33.2 | 16.5 | 11.9 | 1.8 | 3.2 | 10400.1 |
| Power | 6.0 | 7.9 | 0.1 | 0.0 | 4.7 | 1.6 | 1.3 | 0.8 | 0.0 | 0.0 | 3431.0 |
| Industry | 15.8 | 10.3 | 9.1 | 0.4 | 72.9 | 24.6 | 9.3 | 5.8 | 0.6 | 0.6 | 5569.9 |
| Residential | 3.4 | 1.0 | 4.8 | 0.4 | 69.3 | 5.1 | 4.7 | 4.2 | 0.8 | 2.4 | 600.6 |
| Transportation | 0.3 | 8.5 | 5.6 | 0.0 | 29.8 | 0.6 | 0.5 | 0.5 | 0.3 | 0.1 | 849.4 |
| Agriculture | 0.0 | 0.0 | 0.0 | 9.8 | 0.0 | 0.0 | 0.0 | 0.0 | 0.0 | 0.0 | 0.0 |
| Solvent use | 0.0 | 0.0 | 8.6 | 0.0 | 0.0 | 0.0 | 0.0 | 0.0 | 0.0 | 0.0 | 0.0 |
| 2013 | 25.4 | 27.7 | 28.1 | 10.6 | 176.6 | 31.8 | 15.8 | 11.4 | 1.7 | 3.1 | 10450.9 |
| Power | 4.9 | 6.2 | 0.1 | 0.0 | 4.5 | 1.4 | 1.1 | 0.7 | 0.0 | 0.0 | 3359.4 |
| Industry | 12.1 | 10.0 | 9.2 | 0.3 | 65.4 | 20.3 | 8.1 | 5.2 | 0.5 | 0.5 | 5530.3 |
| Residential | 3.1 | 0.9 | 4.5 | 0.4 | 66.7 | 4.8 | 4.4 | 3.9 | 0.8 | 2.2 | 620.1 |
| Transportation | 0.3 | 8.1 | 5.1 | 0.0 | 27.2 | 0.5 | 0.5 | 0.5 | 0.3 | 0.1 | 864.0 |
| Agriculture | 0.0 | 0.0 | 0.0 | 9.8 | 0.0 | 0.0 | 0.0 | 0.0 | 0.0 | 0.0 | 0.0 |
| Solvent use | 0.0 | 0.0 | 10.1 | 0.0 | 0.0 | 0.0 | 0.0 | 0.0 | 0.0 | 0.0 | 0.0 |
| 2014 | 20.4 | 25.3 | 29.1 | 10.5 | 163.8 | 27.0 | 14.1 | 10.3 | 1.6 | 2.8 | 10373.8 |

| | | | | | | | | | | | |
|---|---|---|---|---|---|---|---|---|---|---|---|
| Power | 3.9 | 5.1 | 0.1 | 0.0 | 4.5 | 1.3 | 1.0 | 0.6 | 0.0 | 0.0 | 3318.7 |
| Industry | 9.8 | 9.7 | 9.4 | 0.4 | 56.2 | 15.7 | 6.7 | 4.4 | 0.4 | 0.4 | 5450.0 |
| Residential | 2.9 | 0.9 | 4.2 | 0.4 | 64.0 | 4.4 | 4.1 | 3.6 | 0.7 | 2.0 | 651.5 |
| Transportation | 0.3 | 8.0 | 5.4 | 0.0 | 28.9 | 0.5 | 0.5 | 0.5 | 0.3 | 0.1 | 926.9 |
| Agriculture | 0.0 | 0.0 | 0.0 | 9.7 | 0.0 | 0.0 | 0.0 | 0.0 | 0.0 | 0.0 | 0.0 |
| Solvent use | 0.0 | 0.0 | 9.5 | 0.0 | 0.0 | 0.0 | 0.0 | 0.0 | 0.0 | 0.0 | 0.0 |
| 2015 | 16.9 | 23.7 | 28.5 | 10.5 | 153.6 | 21.9 | 12.3 | 9.1 | 1.5 | 2.5 | 10347.2 |
| Power | 2.7 | 4.6 | 0.1 | 0.0 | 4.6 | 1.3 | 1.0 | 0.6 | 0.0 | 0.0 | 3399.9 |
| Industry | 7.7 | 9.3 | 9.3 | 0.3 | 50.8 | 12.1 | 5.6 | 3.7 | 0.3 | 0.3 | 5290.1 |
| Residential | 2.7 | 0.9 | 3.9 | 0.3 | 60.4 | 4.0 | 3.7 | 3.3 | 0.7 | 1.9 | 661.9 |
| Transportation | 0.3 | 7.7 | 5.0 | 0.0 | 26.2 | 0.5 | 0.5 | 0.5 | 0.3 | 0.1 | 938.8 |
| Agriculture | 0.0 | 0.0 | 0.0 | 9.6 | 0.0 | 0.0 | 0.0 | 0.0 | 0.0 | 0.0 | 0.0 |
| Solvent use | 0.0 | 0.0 | 10.1 | 0.0 | 0.0 | 0.0 | 0.0 | 0.0 | 0.0 | 0.0 | 0.0 |
| 2016 | 13.4 | 22.5 | 28.4 | 10.3 | 141.9 | 17.9 | 10.8 | 8.1 | 1.3 | 2.3 | 10290.6 |
| Power | 1.8 | 4.2 | 0.1 | 0.0 | 4.8 | 1.3 | 1.0 | 0.6 | 0.0 | 0.0 | 3619.2 |
| Industry | 6.0 | 9.2 | 9.7 | 0.3 | 49.2 | 11.1 | 5.2 | 3.5 | 0.3 | 0.3 | 5161.0 |
| Residential | 2.4 | 0.8 | 3.6 | 0.3 | 57.0 | 3.7 | 3.4 | 3.0 | 0.6 | 1.7 | 676.5 |
| Transportation | 0.3 | 7.7 | 4.8 | 0.0 | 25.2 | 0.6 | 0.6 | 0.5 | 0.3 | 0.1 | 977.6 |
| Agriculture | 0.0 | 0.0 | 0.0 | 9.6 | 0.0 | 0.0 | 0.0 | 0.0 | 0.0 | 0.0 | 0.0 |
| Solvent use | 0.0 | 0.0 | 10.4 | 0.0 | 0.0 | 0.0 | 0.0 | 0.0 | 0.0 | 0.0 | 0.0 |
| 2017 | 10.5 | 22.0 | 28.6 | 10.3 | 136.2 | 16.7 | 10.2 | 7.6 | 1.3 | 2.1 | 10434.3 |
| (2013−2010)/2010 | −9% | 5% | 9% | 4% | −5% | −1% | −3% | −4% | 1% | −3% | 16% |
| (2017−2013)/2013 | −59% | −21% | 2% | −3% | −23% | −48% | −36% | −33% | −28% | −32% | 0% |
| (2017−2010)/2010 | −62% | −17% | 11% | 1% | −27% | −48% | −38% | −35% | −27% | −35% | 16% |

[a] The unit of emissions is Tg. [b] TSP is the particulate matter with an aerodynamic diameter of 100 μm or less. [c] $CO_2$ from fossil fuel use and industrial processes.

**Table 2: Comparison of trends in bottom-up emission inventory, satellite-based observations, and top-down emission estimates since 2010.**

| Pollutant | Study | Method[a] | Data | Region[b] | Period | Percent change (%) | Percent change of emissions in this study (%) |
|---|---|---|---|---|---|---|---|
| $SO_2$ | Krotkov et al. (2016) | SAT | OMI $SO_2$ columns | E China | 2010–2015 | −48 | −45 |
| | van der A et al. (2017) | SAT | OMI $SO_2$ columns | China | 2010–2015 | −34 | −39 |
| | Li et al. (2017a) | SAT | OMI $SO_2$ columns | China | 2010–2016 | −68 | −52 |
| | Li et al. (2017a) | IM | $SO_2$ emissions inferred from OMI $SO_2$ columns | China | 2010–2016 | −71 | −52 |
| | Koukouli et al. (2018) | IM | $SO_2$ emissions inferred from OMI $SO_2$ columns | China | 2010–2015 | −27 | −39 |
| $NO_x$ | Krotkov et al. (2016) | SAT | OMI $NO_2$ columns | E China | 2010–2015 | −22 | −14 |
| | Liu et al. (2016) | SAT | OMI $NO_2$ columns | E China | 2010–2015 | −22 | −14 |
| | de Foy et al. (2016) | SAT | OMI $NO_2$ columns | China | 2010–2015 | −12 | −10 |
| | van der A et al. (2017) | IM | $NO_x$ emissions inferred from OMI $NO_2$ columns | E China | 2010–2015 | −8 | −14 |
| | Miyazaki et al. (2017) | IM | $NO_x$ emissions inferred from OMI $NO_2$ columns | China | 2010–2015 | −4 | −10 |
| $NH_3$ | Warner et al. (2017) | SAT | AIRS $NH_3$ VMR[c] | China | 2010–2016 | 9 | 1 |
| CO | Jiang et al. (2017) | IM | CO emissions inferred from MOPITT CO columns | E China | 2010–2015 | −13∼−9 | −18 |
| | Zheng et al. (2018) | IM | CO emissions inferred from MOPITT CO columns | China | 2010–2016 | −25 | −24 |

[a] SAT=satellite-based observations; IM=inverse modeling. [b] E China=Eastern China; [c] VMR=volume mixing ratio.