# Peer review of "Trends in China's anthropogenic emissions since 2010 as the consequence of clean air actions"

_Atmospheric Chemistry and Physics, 2018_

## Referee Comment (RC1) · Anonymous Referee #1 · 4 Jun 2018

This paper is important documentation of the strong air pollution policies in China in the last decade and their consequences on emissions. It comes at the time when new set of scenarios for the IPCC AR6 Report are being finalized and they should take into account these changes, especially for aerosols where climate impacts are or larger significance.

While the estimated emission trends largely coincide with several recent papers reporting observations, the very rapid decline in SO2, especially in the last 2-3 years, appears even stronger here than some of the observations and it is interesting that there seem to be very little (if any) impact on PM2,5 concentrations in the last few years. Of course

no direct translation of SO2 trends to PM2.5 are expected but bearing in mind that apart from NMVOC all other species are reported to either sightly decline or staying constant, it is a a bit of a surprise. I think this deserves a bit more discussion which might bring the issues like regional distribution of changes or stack hight into it. I'd welcome a general discussion not necessarily very detailed one that would probably fit in the section 4.3, which is very short now.

The other element that is not discussed are the uncertainties. There are several elements which are uncertain in the process of estimating emissions and their trends, including the past (not always good) experience in official data reporting and of course the interpretation of remote sensing data, e.g, the quality or ability of monitoring high stack emissions versus low level sources' changes.

I think the paper is well written and has good illustrations. It also includes all key references that i would know of; referring to my comments above I would suggest to add few for the potential discussion (reference to) of particulate matter trends and relation to the emission trends discussed here.

Few more detailed comments:

Page 2, line 1-6: This paragraph includes reference to short lived climate forces and climate, fine, but I'd suggest to review the text and rewrite it slightly as while the authors list PM, ozone and SLCF then in the following impact statement they do not mention regional climate change. It is mentioned later in bold way how they contribute to local and regional ecosystems impacts as well as climate change...but the latter is really CO2 and CH4 in the first place and not pollutants. Yes, SO2 has an important role but its trajectory is not going to fix (tackle) or screw the climate issue.

Page 2, line 11: ' WHO acceptable standards' - be specific to what you refer, I'd suggest changing the wording and say which standard you mean and give reference. Then also the reference in the next sentence to 'this AQ standard' will be clear.

Page 3, line 23: The reference to China (2018); is this including, referring to actual continuous measurement data or an assessment based on the plant operator and regional reports? I think it makes a bit difference in view of the credibility of these. The ref alone does not appear verifiable. Adding few words and certainty and validation of this would be desired.

Page 4, line 1-2: '...covered all emission intensive industries...' To make the statement stronger I'd suggest to add something about embedded enforcement in this regulation and how did it (or not) worked in the past/so far.

Page 4, line 23-24: It is unclear to what is this referring (the economy standards); is this the sticker value given on produced cars or it is real change in the average on the road? My reading would be this is the sticker value for new sold cars and so not necessarily reflecting the real life change at least for two reasons: Real life consumption is somewhere 20-30% higher and in the urban cycle even more, the fleet composition will affect the true impact of such 'sticker' value change. Few words of clarification woudl be useful in the paper.

Page 7, line 8; I am not able to access this http address. The Sliverlight needs to be installed it says but when i try to do it, I get a message that i actually have it (tried on few browsers) and it is not allowed to install again...but effectively i cannot access and view anything from the link. COudl you check please?

Page 8, line 26: I guess it is not only pains and coatings that contribute to strong growth of NMVOC emissions. The whole chemical industry is responsible and there is more to it than just paints. Please verify and adjust if appropriate.

Page 9, line 13 and 18: the authors use words" 'decreased' and 'exhibited' but i'd say rather 'are estimated to decline' ' were estimated ' ... since these are still estimates not entirely free from uncertainties.

Page 9: There is no specific reference to sectors like bricks and coke manufacturing

for which there are no or very few unpublished estimates of actual emissions so how changes/transformation in these sectors included/evaluated? In general the fact that most reductions were estimated to take place in industry, including small industries, the question about monitoring and enforcement arises. It goes without saying that it is harder to monitor progress in policy implementation over 100s thousands sources vs power plant sector for example. I think the paper needs some, even if brief' discussion of this.

Page 10, line 15: 'old vehicles' - I was wondering what happens to them. Are they scrapped or they move to poorer remote provinces? Is there a record of that? Can you add a statement about the fate of these scrapped vehicles? I think this could reinforce the confidence of readers.

Page 10, section 4.3: As mentioned earlier I'd welcome more discussion here, including uncertainty in OMI retrievals, few more words about the studies quotes as SAT or IM in Table 2 as some of them appear to be OMI related studies but you choose to use the IM component of those - something that was not clear to me first. Then there is issue of PM2.5 observations and virtually lak or very small signal visible there - Example of studies where some of the trends are discussed could include: Fei Yao et al (2018; Sci of Tot Env), Fengchao Liang et al (2018, Sco of Tot Env), Rong Xie et al. (2016, Env International), Haifeng Zhang et al (2016, Env Pollution), Tˆania Fontes et al (2017, J. of Env Management), Xiaoyan Wang et al (2018, Amer Met Soc); Li and Sun (2018, A Economy and Space), C.Q. Lin et al (2018, Atm Env). Also in referecne to the above and Figure 8; few more words of explanation there and uncertainties asspociated with it would be very useful. Actually amazing agreement shown here for recent trends (seems certain) while for 2011 strange 'anomaly' ; how well OMI captures changes in emissions of small low level sources like industries or residential coal versus high level stacks - an issue that potentially can lead to overestimation of strong decline in overall emissions.

page 11, section Conclusion; As mentioned earlier, the language of the paper is like

it all was certain but in reality there is a lot of assumptions made and the 'proof' is a mix of reports (not peer reviewed I assume), peer reviewed studies, measurements, and authors assumptions. Some discussion of uncertainty, even if in qualitative terms would be of great value. Again, the refernce and discussion of impacts on the PM2.5 tredsn (all these actions and plans are done for the PM). How sustainable this reduction is, a rebound likely ($CO_2$ in 2016 and 2017 was estimated to show revert trend).

Page 22, Figure 7: I am a bit puzzled about the Figure b where For $SO_2$ only reduction is shown while for other species there is increase from activity driven change. Which sources cause such a change? This is unique to industry it seems, all other charts/sectors show change in the same direction and just the magnitude is different.

---

## Referee Comment (RC2) · Anonymous Referee #2 · 29 Jun 2018

The authors investigated the key trends and drivers of China's anthropogenic emissions for the period of 2010–2017 for the first time. They used a bottom-up emission inventory to quantify emissions for each source sector in each Chinese province, and then combined the estimated emissions data with the Index Decomposition Analysis approach to analyze the drivers of emission trends. The results suggest that China reduced its anthropogenic emissions by a large extent between 2010 and 2017, and emission control measures are the main drivers of this reduction, especially since 2013 when China's Clean Air Action was successfully implemented. The trends in China's emissions are evaluated with both satellite- and ground-based measurement of $SO_2$ and $NO_2$ concentrations, which confirm the certainty of the estimated emissions trends. This work is absolutely within the scope of the ACP journal. Overall, I think the paper reads well, provides valuable results, and could be published after the following issues are addressed. 1. The article makes heavy use of data sets that appear to be confidential or have restricted access, such as the technology penetration data achieved from China's Ministry of Environmental Protection (line 13, page 5). It would be helpful to other researchers if the authors describe these data a bit more clearly, such as which data are used, how these data sources are compiled, and the role these data play in the calculation of emissions in this paper. 2. The emissions trends estimated in this paper are built upon a variety of input data, including official statistics, government reports (not peer reviewed if I understand it correctly), and peer reviewed literatures. I understand the effort made by the authors that update emission inventories to the latest year using a mix of data sources. However, the audience may want to know the certainty of these data and how they affect the certainty of the estimated emissions trends, even if in qualitative terms would be very helpful. For example, care must be taken to confirm that targeted goals/progress from government reports may not be taken as actual emission reductions, although I believe China's emissions are decreasing fast in the last several years after reading this paper. 3. According to the emissions results, China's emissions decreased fast since 2013 mainly due to China's Clean Air Action. I suggest the authors add a bit more description of China's Clean Air Action in the introduction part. Besides, since reducing ambient PM2.5 pollution is the primary objective that stimulate emission control actions, the discussions on PM2.5 concentration trends and the possible linkage to the estimated emission trends may be added in the Sect. 4.3. 4. The authors should be more specific to what they refer in the main text. For example, in line 13 page 3, "to fulfill the air quality target", not clear what the air quality target is. In line 2 page 4, what do the "emission-intensive industries" include? In line 24 page 10, what's the definition of "Eastern China"?

---

## Author Comment (AC1) · 14 Aug 2018

*Reviewer #1:*

*Comments:*

*This paper is important documentation of the strong air pollution policies in China in the last decade and their consequences on emissions. It comes at the time when new set of scenarios for the IPCC AR6 Report are being finalized and they should take into account these changes, especially for aerosols where climate impacts are or larger significance.*

**Response:**

We thank the reviewer #1 for the constructive comments and address them as below.

*While the estimated emission trends largely coincide with several recent papers reporting observations, the very rapid decline in SO₂, especially in the last 2-3 years, appears even stronger here than some of the observations and it is interesting that there seem to be very little (if any) impact on PM₂.₅ concentrations in the last few years. Of course no direct translation of SO₂ trends to PM₂.₅ are expected but bearing in mind that apart from NMVOC all other species are reported to either slightly decline or staying constant, it is a bit of a surprise. I think this deserves a bit more discussion which might bring the issues like regional distribution of changes or stack height into it. I'd welcome a general discussion not necessarily very detailed one that would probably fit in the section 4.3, which is very short now.*

**Response:**

Satellite-derived $PM_{2.5}$ concentrations were flat over China during 2010 and 2013 (Fig. 8a), corresponding to small variations in emissions of different precursors estimated for the same period. Satellite-based $PM_{2.5}$ concentrations decreased by 18% from 2013–2015, in good agreement with trend in surface $PM_{2.5}$ concentrations over 74 cities in China. During 2013–2017, surface $PM_{2.5}$ concentrations over 74 cities decreased by 35%. We estimated faster decrease in $SO_2$ emissions (−59%) than observed surface $PM_{2.5}$ concentrations, while the estimated decrease rates of $NO_x$ (−21%) and $NH_3$ (−3%) emissions were slower than observed $PM_{2.5}$ concentrations. This phenomenon was qualitatively confirmed by observed large decrease of sulfate and increased relative contribution of nitrate and ammonium in $PM_{2.5}$ compositions from 2013–2017 (Shao et al., 2018). We added a discussion on the trend of $PM_{2.5}$ concentrations and the relation to estimated emissions in Sect. 4.3 as the reviewer suggested.

*The other element that is not discussed are the uncertainties. There are several elements which are uncertain in the process of estimating emissions and their trends, including the past (not always good) experience in official data reporting and of course the interpretation of remote sensing data, e.g, the quality or ability of monitoring high stack emissions versus low level sources' changes.*

**Response:**

We add a discussion in Sect. 4.3 to discuss the uncertainties in OMI retrievals and in emission estimates, as well as their influence on comparison between emission trends and observations. Please refer to Sect. 4.3 in the revised manuscript for more details.

*I think the paper is well written and has good illustrations. It also includes all key references that I would know of; referring to my comments above I would suggest to add few for the potential discussion (reference to) of particulate matter trends and relation to the emission trends discussed here.*

**Response:**

We add a discussion on the trend of $PM_{2.5}$ concentrations and the relation to estimated emissions in Sect. 4.3. Please refer to the response to the 1st comment.

*Few more detailed comments:*

*Page 2, line 1-6: This paragraph includes reference to short lived climate forces and climate, fine, but I'd suggest to review the text and rewrite it slightly as while the authors list PM, ozone and SLCF then in the following impact statement they do not mention regional climate change. It is mentioned later in bold way how they contribute to local and regional ecosystems impacts as well as climate change...but the latter is really $CO_2$ and $CH_4$ in the first place and not pollutants. Yes, $SO_2$ has an important role but its trajectory is not going to fix (tackle) or screw the climate issue.*

**Response:**

We rewrite this paragraph to add the statement of regional climate change as follows.

"These pollutants constitute the majority of the precursors of $PM_{2.5}$ and $O_3$ pollution as well as those of short-lived climate forcers, which exert harmful effects on human health, agriculture, and regional climate. These pollutants not only cause local to regional environmental problems such as premature deaths and agricultural yield losses, but also have significant impact on regional climate changes in temperature and precipitation. To tackle the problems of both air pollution and regional climate change, it is important to fully understand the trends and drivers of Chinese emissions."

*Page 2, line 11: ' WHO acceptable standards' - be specific to what you refer, I'd suggest changing the wording and say which standard you mean and give reference. Then also the reference in the next sentence to 'this AQ standard' will be clear.*

**Response:**

Corrected.

*Page 3, line 23: The reference to China (2018); is this including, referring to actual continuous measurement data or an assessment based on the plant operator and regional reports? I think it*

*makes a bit difference in view of the credibility of these. The ref alone does not appear verifiable. Adding few words and certainty and validation of this would be desired.*

**Response:**

The reference to China (2018) suggests that 71% of installed capacity of power plants operated close to "ultralow emission" levels in 2017. This figure is estimated on the basis of firm-level information of pollution control devices and efficiencies, which are collected from each plant by local agencies, and then managed and verified by Ministration of Environmental Protection of China. The power plants that comply with "ultralow emission" standards are mainly large ones at the current stage. Most of them use continuous emission monitoring systems to monitor exhaust emissions, which confirm that these plants are indeed complying with the "ultralow" emission standards. We add a discussion on the credibility of China (2018) to make the statement stronger.

*Page 4, line 1-2: '...covered all emission intensive industries...' To make the statement stronger I'd suggest to add something about embedded enforcement in this regulation and how did it (or not) worked in the past/so far.*

**Response:**

We now added examples of cement plants and industrial boilers to this paragraph to illustrate the enforcement of emission limits. The emission limits of cement plants were 800 mg m$^{-3}$ for NO$_x$ and 50 mg m$^{-3}$ for particulates before 2014 (the standard GB 4915-2004), while after 2014 all cement plants were required to reach new limit values of 400 mg m$^{-3}$ for NO$_x$ and of 30 mg m$^{-3}$ for particulates (the standard GB 4915-2013). For coal boilers used in all of the types of industries, the emission limits were 900 mg m$^{-3}$ for SO$_2$ and 80–250 mg m$^{-3}$ for particulates before 2014 (the standard GB 13271-2001), and no limits were required for NO$_x$. After 2014, new coal-fired industrial boilers faced stricter limit values of 300, 300 and 50 mg m$^{-3}$ for SO$_2$, NO$_x$ and particulates (the standard GB 13271-2014), respectively. The introduction of new emission standard in 2014 also tightened limit values for the existing coal-fired industrial boilers, where the "not to exceed" limits for SO$_2$, NO$_x$ and particulates were 400, 400 and 80 mg m$^{-3}$, respectively.

*Page 4, line 23-24: It is unclear to what is this referring (the economy standards); is this the sticker value given on produced cars or it is real change in the average on the road? My reading would be this is the sticker value for new sold cars and so not necessarily reflecting the real life change at least for two reasons: Real life consumption is somewhere 20-30% higher and in the urban cycle even more, the fleet composition will affect the true impact of such 'sticker' value change. Few words of clarification would be useful in the paper.*

**Response:**

True. China's economy standards refer to the fuel consumption rates of vehicles tested under the European standard driving cycle in laboratory. The tested fuel efficiency are shown on fuel economy labels (window stickers) of new sold cars. The real-world fuel consumption rates are typically 15% higher than these sticker values (Huo et al., 2011), because the European test procedure cannot reflect the real urban and highway driving conditions in China. We clarify these in the revised manuscript.

*Page 7, line 8; I am not able to access this http address. The Sliverlight needs to be installed it says but when I try to do it, I get a message that I actually have it (tried on few browsers) and it is not allowed to install again...but effectively I cannot access and view anything from the link. Could you check please?*

**Response:**

I can access the URL of http://106.37.208.233:20035/ using Internet Explorer. I tried two different computers (Windows 7 system) and accessed this website after installing Silverlight. If that doesn't work for you, you can also view the archived observational data at the website of http://beijingair.sinaapp.com/.

*Page 8, line 26: I guess it is not only paints and coatings that contribute to strong growth of NMVOC emissions. The whole chemical industry is responsible and there is more to it than just paints. Please verify and adjust if appropriate.*

**Response:**

NMVOC emissions from paints and coatings increased by 2.4 Tg from 2010–2017, which are the largest contributor to the growth of 2.7 Tg emissions from all source sectors. The strong growth of paints can be attributed to the increasing demand to coat buildings, cars, and machinery due to the rapid increase in the area of newly built house (+52%) and the production of vehicles (+54%). Chemical industry increased 1.5 Tg NMVOC emissions from 2010–2017, making them the second largest contributor to NMVOC growth. We clarify this in the manuscript.

*Page 9, line 13 and 18: the authors use words" 'decreased' and 'exhibited' but I'd say rather 'are estimated to decline' ' were estimated ' ... since these are still estimates not entirely free from uncertainties.*

**Response:**

Corrected.

*Page 9: There is no specific reference to sectors like bricks and coke manufacturing for which there are no or very few unpublished estimates of actual emissions so how changes/transformation in these sectors included/evaluated? In general the fact that most reductions were estimated to take place in industry, including small industries, the question about monitoring and enforcement arises. It goes without saying that it is harder to monitor progress in policy implementation over 100s thousands sources vs power plant sector for example. I think the paper needs some, even if brief' discussion of this.*

**Response:**

Brick and coke manufacturing industries have seen strict emissions standards since 2010 (Fig. 1), and pollutants generated by these regulated sources are monitored and managed by local agencies.

Each province submits annual implementation report to China's Ministry of Environmental Protection to summarize the progress in pollution control every year, and we derived the information of emission changes from those reports. These are the best data sources available now, but we still agree that the statistics for thousands of small industrial sources tend to be more uncertain than the large industries that have good record in pollution levels. We summarize these information in Sect. 4.3, and discuss the difficulty to monitor progress in pollution control over small industries and its influence on uncertainties of emission trend estimates.

*Page 10, line 15: 'old vehicles' - I was wondering what happens to them. Are they scrapped or they move to poorer remote provinces? Is there a record of that? Can you add a statement about the fate of these scrapped vehicles? I think this could reinforce the confidence of readers.*

**Response:**

China has scrapped all the old vehicles that don't meet stringent emission standards, i.e., "yellow label" vehicles, by the end of 2017. The number of vehicles scrapped in each province are recorded by local government, and these scrapped vehicles are banned from roads and sent to wrecking yard for recycling. We clarify this in the revised manuscript.

*Page 10, section 4.3: As mentioned earlier I'd welcome more discussion here, including uncertainty in OMI retrievals, few more words about the studies quotes as SAT or IM in Table 2 as some of them appear to be OMI related studies but you choose to use the IM component of those - something that was not clear to me first. Then there is issue of PM2.5 observations and virtually lak or very small signal visible there - Example of studies where some of the trends are discussed could include: Fei Yao et al (2018; Sci of Tot Env), Fengchao Liang et al (2018, Sco of Tot Env), Rong Xie et al. (2016, Env International), Haifeng Zhang et al (2016, Env Pollution), Tania Fontes et al (2017, J. of Env Management), Xiaoyan Wang et al (2018, Amer Met Soc); Li and Sun (2018, A Economy and Space), C.Q. Lin et al (2018, Atm Env). Also in reference to the above and Figure 8; few more words of explanation there and uncertainties associated with it would be very useful. Actually amazing agreement shown here for recent trends (seems certain) while for 2011 strange 'anomaly' ; how well OMI captures changes in emissions of small low level sources like industries or residential coal versus high level stacks - an issue that potentially can lead to overestimation of strong decline in overall emissions.*

**Response:**

For the IM studies in Table 2, we clarify the satellite observations they used to constrain emissions, including OMI columns of $SO_2$ and $NO_2$ and MOPITT CO columns.

We add a discussion on the trend of $PM_{2.5}$ concentrations and their relation to the estimated emissions in Sect. 4.3. The correlation between $PM_{2.5}$ concentrations and emissions of $PM_{2.5}$ precursors are analyzed, and the associated uncertainties are discussed. For more details please refer to the revised manuscript.

The uncertainty in using OMI retrievals to infer emissions is also discussed in Sect. 4.3. Interannual variabilities can result in remarkable variations in column concentrations (Uno et al., 2007), which may partly explain the disagreement between changes in emissions and observations for a signal

year (e.g., year 2011 in Fig. 8b). In addition, satellite-based column observations are typically more visible to high-stack emissions. For example, $SO_2$ columns are less sensitive to small and near surface emissions (Li et al., 2017), which may lead to an underestimation of $SO_2$ budget using satellite data in China for most recent years and a disagreement between emission and $SO_2$ column trend when high-stack emissions (e.g., power plants) were significantly reduced.

*page 11, section Conclusion; As mentioned earlier, the language of the paper is like it all was certain but in reality there is a lot of assumptions made and the 'proof' is a mix of reports (not peer reviewed I assume), peer reviewed studies, measurements, and authors assumptions. Some discussion of uncertainty, even if in qualitative terms would be of great value. Again, the reference and discussion of impacts on the $PM_{2.5}$ trends (all these actions and plans are done for the PM). How sustainable this reduction is, a rebound likely ($CO_2$ in 2016 and 2017 was estimated to show revert trend).*

**Response:**

In the revised manuscript, we use the words like "were estimated to decline" to clarify that the conclusions are made based on our bottom-up emission estimates. We also summarize the uncertainties of emission estimates in qualitative terms in Sect. 4.3.

For the $PM_{2.5}$ trends, we add a sentence in the conclusion section as "the emissions trends of $PM_{2.5}$ precursors agree well with changes in $PM_{2.5}$ compositions over China". Detailed discussions are provided in Sect. 4.3 as the reviewer suggested.

We think the reduction in China's air pollutant emissions is very unlikely to rebound for the following reason. All the reductions in emissions from 2010–2017 were driven by the objective to reduce $PM_{2.5}$ pollutions in China. The Clean Air Action implemented since 2013 has cut annual average $PM_{2.5}$ concentrations by 35% during the period of 2013–2017. For years after 2017, all cities that exceed the 35 $\mu g\ m^{-3}$ annual standard are further required to reduce annual average $PM_{2.5}$ concentrations by 18% below the 2015 level in 2020. Since the annual average limit of $PM_{2.5}$ is exceeded in many Chinese cities currently, the 2020 air quality target will continue driving down China's air pollutant emissions in the future. We clarify this in the conclusion section.

*Page 22, Figure 7: I am a bit puzzled about the Figure b where For $SO_2$ only reduction is shown while for other species there is increase from activity driven change. Which sources cause such a change? This is unique to industry it seems, all other charts/sectors show change in the same direction and just the magnitude is different.*

**Response:**

For the industry sector, the activity driven decrease in $SO_2$ emissions is caused by reduced coal use in industrial boilers. This also reduces emissions of all the other species in the industry sector. However, for other emission species, the activity increase driven by other industrial sources totally offset the effect of decreasing activities from industrial boilers. For example, the iron and steel industry drives up CO emissions; the cement industry drives up $NO_x$ emissions; coke, iron, and steel industries drive up particulate matter emissions. This phenomenon is only observed in the industry sector, because this sector is a combination of many industrial sources in this study. The

total effect of activity and pollution control on industrial emissions need to consider all the detailed sources as well as their emission shares in the industry sector.

Reference

Huo, H., Yao, Z., He, K., and Yu, X.: Fuel consumption rates of passenger cars in China: Labels versus real-world, Energy Policy, 39, 7130-7135, doi: 10.1016/j.enpol.2011.08.031, 2011.

Li, C., McLinden, C., Fioletov, V., Krotkov, N., Carn, S., Joiner, J., Streets, D., He, H., Ren, X., Li, Z., and Dickerson, R. R.: India Is Overtaking China as the World's Largest Emitter of Anthropogenic Sulfur Dioxide, Scientific Reports, 7, 14304, doi:10.1038/s41598-017-14639-8, 2017.

Shao, P., Tian, H., Sun, Y., Liu, H., Wu, B., Liu, S., Liu, X., Wu, Y., Liang, W., Wang, Y., Gao, J., Xue, Y., Bai, X., Liu, W., Lin, S., and Hu, G.: Characterizing remarkable changes of severe haze events and chemical compositions in multi-size airborne particles ($PM_1$, $PM_{2.5}$ and $PM_{10}$) from January 2013 to 2016–2017 winter in Beijing, China, Atmos. Environ., 189, 133-144, doi: 10.1016/j.atmosenv.2018.06.038, 2018.

Uno, I., He, Y., Ohara, T., Yamaji, K., Kurokawa, J. I., Katayama, M., Wang, Z., Noguchi, K., Hayashida, S., Richter, A., and Burrows, J. P.: Systematic analysis of interannual and seasonal variations of model-simulated tropospheric $NO_2$ in Asia and comparison with GOME-satellite data, Atmos. Chem. Phys., 7, 1671-1681, doi: 10.5194/acp-7-1671-2007, 2007.

---

## Author Comment (AC2) · 14 Aug 2018

*Reviewer #2:*

*Comments:*

*The authors investigated the key trends and drivers of China's anthropogenic emissions for the period of 2010–2017 for the first time. They used a bottom-up emission inventory to quantify emissions for each source sector in each Chinese province, and then combined the estimated emissions data with the Index Decomposition Analysis approach to analyze the drivers of emission trends. The results suggest that China reduced its anthropogenic emissions by a large extent between 2010 and 2017, and emission control measures are the main drivers of this reduction, especially since 2013 when China's Clean Air Action was successfully implemented. The trends in China's emissions are evaluated with both satellite- and ground-based measurement of SO$_2$ and NO$_2$ concentrations, which confirm the certainty of the estimated emissions trends. This work is absolutely within the scope of the ACP journal. Overall, I think the paper reads well, provides valuable results, and could be published after the following issues are addressed.*

**Response:**

We thank the reviewer #2 for the comments and our point-by-point response is given below.

*1. The article makes heavy use of data sets that appear to be confidential or have restricted access, such as the technology penetration data achieved from China's Ministry of Environmental Protection (line 13, page 5). It would be helpful to other researchers if the authors describe these data a bit more clearly, such as which data are used, how these data sources are compiled, and the role these data play in the calculation of emissions in this paper.*

**Response:**

We obtained the firm-level statistics for electric generators, cement factories, iron- and steel-making furnaces, and glass kilns from China's Ministration of Environmental Protection (MEP). These data are collected from each plant by local agencies, and then managed and verified by MEP. The information adopted in this study include pollution control technologies, penetrations, and efficiencies for different industries in each province, which are used to calibrate emission control levels (i.e., *C* and *η* in Eq. (1)) in the bottom-up inventory. We clarify this in Sect. 3.1.

*2. The emissions trends estimated in this paper are built upon a variety of input data, including official statistics, government reports (not peer reviewed if I understand it correctly), and peer reviewed literatures. I understand the effort made by the authors that update emission inventories to the latest year using a mix of data sources. However, the audience may want to know the certainty of these data and how they affect the certainty of the estimated emissions trends, even if in qualitative terms would be very helpful. For example, care must be taken to confirm that targeted goals/progress from government reports may not be taken as actual emission reductions, although I believe China's emissions are decreasing fast in the last several years after reading this paper.*

**Response:**

We add a discussion on the uncertainties of emission trend estimates in qualitative terms as the reviewer suggested. Please refer to Sect. 4.3 for more details.

*3. According to the emissions results, China's emissions decreased fast since 2013 mainly due to China's Clean Air Action. I suggest the authors add a bit more description of China's Clean Air Action in the introduction part. Besides, since reducing ambient PM2.5 pollution is the primary objective that stimulate emission control actions, the discussions on PM2.5 concentration trends and the possible linkage to the estimated emission trends may be added in the Sect. 4.3.*

**Response:**

We add the following sentences in the introduction part to describe China's Clean Air Action.

"The Clean Air Action is China's first five year plan (2013–2017) that radically tightened air pollution targets for particulate matter pollution reduction. The three metropolitan regions mentioned above were required to reduce $PM_{2.5}$ concentrations by 15–25% by the year 2017 compared with the 2013 levels, and all other provinces in China were required to reduce $PM_{10}$ concentrations by 10%. The Clean Air Action launched stringent measures to achieve these air quality targets, including the adjustment of energy mix and industrial structure, reduction of air pollutant emissions, establishment of monitoring and early-warning systems for air pollution, and other supportive policies. With the successful policy implementation, China met the 2017 air pollution target set under 2013 Clean Air Action, and the annual average $PM_{2.5}$ concentrations were reduced by 28–40% from 2013–2017 in the three metropolitan regions (China, 2018)."

We add a general discussion on $PM_{2.5}$ concentration trends and the relation to the estimated emissions in the first paragraph of Sect. 4.3.

*4. The authors should be more specific to what they refer in the main text. For example, in line 13 page 3, "to fulfill the air quality target", not clear what the air quality target is. In line 2 page 4, what do the "emission-intensive industries" include? In line 24 page 10, what's the definition of "Eastern China"?*

**Response:**

The air quality targets refer to those set under 2013 Clean Air Action, which are described in detail in the introduction part now. We change the sentence to "To fulfill the air quality target set under 2013 Clean Air Action".

The emission-intensive industries mainly include iron and steel making, cement, brick, coke, glass, and chemical industries. We clarify this in the main text.

The Eastern China discussed in this paper includes the provinces of Beijing, Tianjin, Hebei, Shanxi, Shaanxi, Shandong, Henan, Hubei, Anhui, Jiangsu, Shanghai, and Zhejiang. The definition of Eastern China is given in the caption of Fig. 8.

---

## Author Response (AR2)

*Minor comments*

*Page 4, line 24: do you mean 'Phase out small and polluted factories'? Is 'Phase out small, high emitting factories' better?*
*5, 15: delete 'stringent' or make it 'more stringent'.*
*8, 1: '… are estimated to have declined by…'*
*8, 8: '… the more stringent pollution…'*
*8, 15: '… are estimated to have increased by…'*
*9, 2: '… are estimated to have increased by…'*
*9, 28: '… After 2013, emissions of all air pollutants except… are found to have reduced as a result of pollution controls.'*
*10, 3: '… are estimated to have decreased by…'*
*11, 6: '…that don't meet the more stringent emission…'*
*12, 3: 'We estimate a faster decrease…' – though please re-read this sentence to make it as clear as possible.*
*12, 10: '.. rapid decreases of..'*
*12, 11: '..China decreased by…'*
*12, 14/15: '…agreement with the estimated emission trend. In contrast, surface…'*
*12, 29: '…more accurate. However, surface….'*
*12, 33/34: '…are compared. However, influences….'*
*13, 1: '…due to the large..'*
*13, 9: 'Many of the uncertainties…'*
*13, 10: '.. have less impact on emission trends (Lu et al., 2011), but non-compliance with…'*
*14, 4: '… PM2.5 concentrations to 18% below…' – please check this what you mean.*

**Response:**

We thank the editor for the comments and have revised our manuscript as the editor suggested.

[revised manuscript text omitted]
 PM2.5 precursors in a. The 2010–2017 trends in SO2 (red, the solid curve in b) and NOx (blue, the solid curve in b) emissions are compared with OMI SO2 (red, the dashed curve in b) and NO2 (blue, the dashed curve in b) tropospheric columns for Eastern China, respectively. Eastern China here includes the provinces of Beijing, Tianjin, Hebei, Shanxi, Shaanxi, Shandong, Henan, Hubei, Anhui, Jiangsu, Shanghai, and Zhejiang. Besides, the 2013–2017 trends in ground-based observations of SO2 (red, the dotted curve in b), NO2 (blue, the dotted curve in b), and PM2.5 (black, the dotted curve in a) are also presented.

[revised manuscript text omitted]